# INTERPRETING AND EDITING VISION-LANGUAGE REPRESENTATIONS TO MITIGATE HALLUCINATIONS

**Nick Jiang**[*], **Anish Kachinthaya**[*], **Suzie Petyrk**[†],**Yossi Gandelsman**[†]
University of California, Berkeley
{nickj,anishk,spetryk,yossi_gandelsman}@berkeley.edu

## ABSTRACT

We investigate the internal representations of vision-language models (VLMs) to address hallucinations, a persistent challenge despite advances in model size and training. We project VLMs' internal image representations to their language vocabulary and observe more confident output probabilities on real objects than hallucinated objects. We additionally use these output probabilities to spatially localize real objects. Building on this approach, we introduce a knowledge erasure algorithm that removes hallucinations by linearly orthogonalizing image features with respect to hallucinated object features. We show that targeted edits to a model's latent representations can reduce hallucinations by up to 25.7% on the COCO2014 dataset while preserving performance. Our findings demonstrate how a deeper understanding of VLMs' latent representations can enhance reliability and enable novel capabilities, such as zero-shot segmentation.[1]

## 1 INTRODUCTION

Vision-Language Models (VLMs) have recently emerged as powerful tools for understanding images via text (Dai et al., 2023; Liu et al., 2024a). They have demonstrated remarkable capabilities across multimodal tasks such as image captioning (Li et al., 2023a), visual question answering (Ye et al., 2023), and complex multimodal reasoning (Bai et al., 2023). Despite their capabilities, VLMs tend to hallucinate content that does not appear in the images (Ji et al., 2023), which poses serious concerns for the reliability of these models in real-world applications (Hu et al., 2023; Luo et al., 2024).

Widespread belief has been that scaling to larger models and more training data will naturally mitigate hallucinations. However, recent studies have shown that hallucinations persist even in larger and more advanced models (Rohrbach et al., 2019; Li et al., 2023b), suggesting that this issue cannot be solved by scale alone. Current methods reduce hallucinations by applying external interventions (e.g. object detectors; Yin et al. (2023)) or additional model fine-tuning (e.g. on hallucination examples; Zhou et al. (2024); Zhang et al. (2024a)). Nevertheless, these methods often struggle to distinguish between subtle hallucinations and existing details, requiring new models or updated model parameters.

In this paper, we aim to introduce fine-grained edits directly to the image latent representations of VLMs to reduce hallucinations without hindering their performance, an approach that has had some success in large language models (Zhang et al., 2024b; von Rutte et al., 2024). To edit the latent representations of VLMs, we first explain their role via text. We employ the logit lens technique (nostalgebraist, 2020) to directly interpret the spatial VLM *image* representations with VLM *text vocabulary*. Surprisingly, the characteristics of these image representations are different for real objects that appear in the image and objects that are hallucinated. Moreover, the logit lens enables spatially localizing objects within the input image.

Relying on the ability to detect hallucinated objects, we edit them out by intervening in their internal representations. We introduce a knowledge erasure algorithm, PROJECTAWAY, to target and remove objects by linearly orthogonalizing image features with respect to the text features of target objects. We find that PROJECTAWAY can erase both real and hallucinated objects with high rates of removal.

---

[*]Equal contribution as first authors.
[†]Equal contribution as last authors.
[1]Code: https://github.com/nickjiang2378/vl-interp

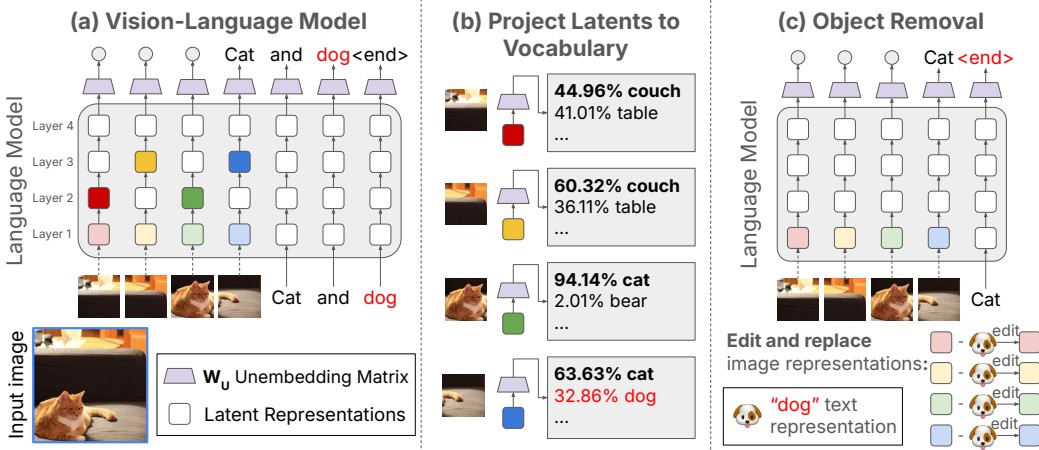

Figure 1: **Interpreting VLM internal image representations**. (a) Given a VLM, (b) we unembed the latent representations from image embeddings to the vocabulary and classify hallucinations. We remove hallucinations by (c) linearly editing them out of the latent representations.

We use our interpretation and editing approach for three tasks. First, we utilize the logit lens on image features to detect hallucinations in the image. We find that it improves mAP by 22.45% and 47.17% in two VLMs. Then, we combine our editing and detection method to erase hallucinations from the VLM's internal representations, reducing hallucinations up to 25.7% on standard benchmarks, while preserving accuracy in image captioning. Finally, we use the logit lens to localize objects in the image features. We find that our spatial mapping provides comparable performance to state-of-the-art zero-shot segmentation methods. Our results indicate that understanding the internal representations of VLMs can be achieved and used to repair model hallucinations and introduce new capabilities.

## 2 RELATED WORK

### 2.1 INTERPRETING LATENT REPRESENTATIONS IN LANGUAGE MODELS

Interpreting the inner workings of large language models enables fine-grained improvement of the language model behavior. Recent work involves utilizing the model's attention maps (Kobayashi et al., 2020; Chefer et al., 2021), activation patterns (Conmy et al., 2023; Meng et al., 2023; Bronzini et al., 2024), and latent representations (Ghandeharioun et al., 2024; Cunningham et al., 2023; Bricken et al., 2023) to understand their behavior with applications such as early exiting (Halawi et al., 2024) and editing or erasing the model's knowledge (Dai et al., 2022; Ravfogel et al., 2024). One class of methods probe the VLMs knowledge with linear classifiers (Hewitt & Manning, 2019; Tucker et al., 2021; Li et al., 2024; Belrose et al., 2023). The logit lens method (nostalgebraist, 2020), which we will use in our analysis, finds the output distribution over the vocabulary of the language model at intermediate layers with the model's own unembedding matrix. We apply this approach to *VLMs* to interpret the model's understanding of *visual information* in the model's textual vocabulary.

### 2.2 INTERPRETING LATENT REPRESENTATIONS IN VISION MODELS

Understanding the internal dynamics of vision models is critical for ensuring safety and reliability in multimodal systems. Early works in this area focused on producing saliency maps (Petsiuk et al., 2018), analyzing individual neurons (Bau et al., 2020; 2019; Dravid et al., 2023), and training networks to map latent representations to concepts (Esser et al., 2020). With the emergence of transformer-based vision models like CLIP (Radford et al., 2021), recent methods explain latent tokens (Chen et al., 2023) and the roles of attention heads and neurons with natural language (Gandelsman et al., 2024b;a). Few works currently interpret the internal computation of VLMs: Palit et al. (2023) develop a neuron causal tracing tool; Schwettmann et al. (2023) identifies multi-modal neurons; and Huo et al. (2024) ablates domain-specific neurons to improve vision question-answering. Whereas past

papers have primarily studied the mechanisms (e.g. neuron analysis) that drive VLMs, we focus on interpreting and editing their latent representations for real-world applicability.

## 2.3 DETECTING AND REDUCING VLM HALLUCINATIONS

While VLM performances on image caption and visual question answering are continually improving, they continue to hallucinate facts that are not supported by the visual input. Existing methods for detecting hallucinations in language models during inference utilize latent representations (He et al., 2024; Su et al., 2024), activations (Chen et al., 2024), and output logit values (Varshney et al., 2023). SAPLMA (Azaria & Mitchell, 2023) trains a hallucination classifier on the internal latent representations. LUNA (Song et al., 2024) learns a transition function on latent representations and identifies abnormal transitions. Varshney et al. (2023) uses the final layer logits to score the model's confidence in an entity or keyword and intervenes by instructing the model to either repair or remove the hallucinated information. Among VLMs, LURE (Zhou et al., 2024) is a fine-tuned revisor model to detect and reduce hallucinations. OPERA (Huang et al., 2024) uses the model's internal attention weights to detect and suppress patterns that align with the beginning of hallucinated phrases. In contrast to these methods, we leverage the internal *image* representations in the VLMs for hallucination reduction and for zero-shot segmentation.

## 3 EXTRACTING KNOWLEDGE FROM VLMS

We start by introducing VLMs and the general framework of their architectures in most recent work. We then describe our approach for decoding the features in intermediate image representations in VLMs into text, and apply it to two types of VLMs. Surprisingly, this approach effectively probes the knowledge about objects present in images and can localize objects within the image.

### 3.1 PRELIMINARIES

**Vision-Language Models.** The architecture of recent state-of-the-art VLMs for text generation typically involves three main components: a vision encoder to process image inputs, a mapping network to map image features to image embeddings, and an autoregressive language model to process the image embeddings and prompt embeddings to generate text. We focus on two recent state-of-the-art VLMs: *LLaVA* 1.5 (Liu et al., 2024a) and *InstructBLIP* (Dai et al., 2023). We use 7B versions of both these models. LLaVA utilizes a frozen CLIP vision encoder and an MLP as a mapping network to project the vision encoder outputs into image embeddings for the language model. The MLP is pre-trained on a large vision-language dataset and both the MLP and the language model are fine-tuned on an instruction-focused dataset. In contrast, InstructBLIP freezes both the vision encoder and the language model and only trains the mapping network.

**Notations.** For the purposes of our work, we define the VLM architecture as follows. The vision encoder processes an input image to produce $n$ image features. These image features are projected to embedding space via the mapping network, resulting in $n$ $d$-dimensional image embeddings $\{k_i : k_i \in \mathbb{R}^d, i = 1, ..., n\}$. For the language model, the entire set of text tokens constitutes the vocabulary $V$ with vocabulary size $|V|$. The image embeddings, followed by $m$ text embeddings $\{t_i : t_i \in \mathbb{R}^d, i = 1, ..., m\}$ of the prompt tokens, are input to the language model through $L$ decoder layers. For an input embedding $x \in \mathbb{R}^d$, we define $h_l(x) \in \mathbb{R}^d$ to be the latent representation for embedding $x$ at layer $l \in \{1, ..., L\}$, the output of the decoder layer, which is conditioned on previous tokens of the input sequence. An unembedding matrix $W_U \in \mathbb{R}^{|V| \times d}$ maps the last latent representation $h_L(t_m)$ to a probability distribution over the vocabulary for the next token $t_{m+1}$.

**Logit Lens.** Logit Lens is an interpretability method for intermediate language model representations introduced in Section 2.1. The logit lens technique applies the unembedding matrix $W_U$ to latent representations $h_l(x)$ in the $L$ intermediate layers in the language model to retrieve the logit distributions over the vocabulary.

$$f_l(t_m) = W_U \cdot h_l(t_m) = [\text{logit}_1, \text{logit}_2, \text{logit}_3, \ldots, \text{logit}_{|V|}] \tag{1}$$

This is the logit distribution representing the predictions of the model after $l$ layers, where $\text{logit}_j$ corresponds to the token $j$ in the vocabulary.

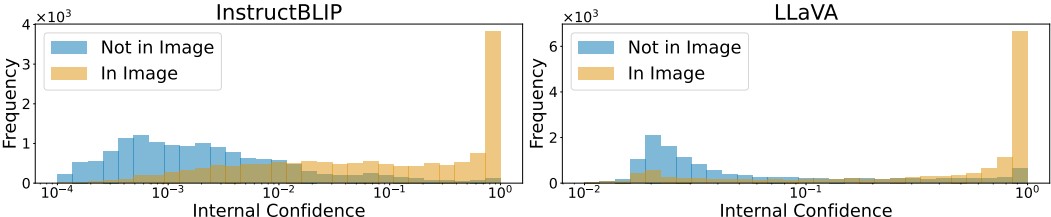

Figure 2: **Comparison of internal confidence in objects present and not present in the image**. We examine the internal confidence of COCO objects that exist and do not exist in the image within intermediate VLM image representations. We observe that objects that do not exist in the image have lower internal confidence.

## 3.2 APPLYING LOGIT LENS ON VLMS

We apply the logit lens to probe the language model as it processes the image representations. This enables us to interpret the image features' output distributions as they are transformed by the layers of the language model and localize objects spatially within the image.

**Extracting probability distributions from intermediate image representations**. We apply logit lens on the *image representations* in the VLM. For a given image embedding $k_i$, we find the latent representation of the image embedding at layer $l$, $h_l(k_i)$, taking the logit lens to get the probability distribution over the vocabulary, softmax$(f_l(k_i))$. We define an object $o$, an object word composed of tokens from the vocabulary. We inspect the probability of a specific object $o$, softmax$(f_l(k_i))_o$. For multi-token objects, we take the maximum probability value over the object tokens. This provides a generalizable framework for analyzing specific latent image representations via text, with respect to specific objects. Next, we find the maximum probability over all image representations over all layers. For object $o$, we compute:

$$c_o = \max_{\substack{1 \leq l \leq L \\ 1 \leq i \leq n}} \left\{ \text{softmax}(f_l(k_i))_o \right\} \tag{2}$$

We define $c_o$ as the VLMs *internal confidence* of an object $o$ existing in the image: the highest probability of object presence across $n$ image representations through $L$ layers of the language model.

**Comparing the internal confidence of present and not present objects.** To determine if internal confidence provides meaningful information about objects in the image, we examine $c_o$ for objects present and not present in an image. We use InstructBLIP and LLaVA to caption 5000 random COCO2014 images in the Karpathy validation split (Lin et al., 2015) and determine $c_o$ for all 80 COCO objects, only a few of which are present in each image. Since there are many more objects not present than present, we randomly sample a subset of the internal confidences for objects not present. Figure 2 exhibits the internal confidences for objects present and not present in the image. We empirically find that the VLMs' internal confidences are higher for present objects than not present ones. We use this claim later to classify objects as hallucinations in Section 5.1.

**Object localization**. Given that the language model can distinguish between objects present and not present in an image, we examine whether it can attribute high object internal confidence to specific patches in an image. For each image embedding $k_i$ in $n$ image embeddings, we find the maximum softmax probability of an object within the layers of the model, $\max_{1 \leq l \leq L}\{\text{softmax}(f_l(k_i))_o\}$. Using these internal confidence values, we localize the objects in the image patches, each of which maps to an image embedding. We focus on LLaVA for this task, since its image encoder preserves the spatial mapping of image patches to image features.

We observe that image representations that exhibit higher internal confidence for specific objects correspond to the image patches in which those objects are visually present (examples in Figure 3). Building on our previous observation, we see that the intermediate image representations semantically align with latent token representations of objects present in them while maintaining their spatial locality. We use this unique finding for zero-shot segmentation in Section 5.3.

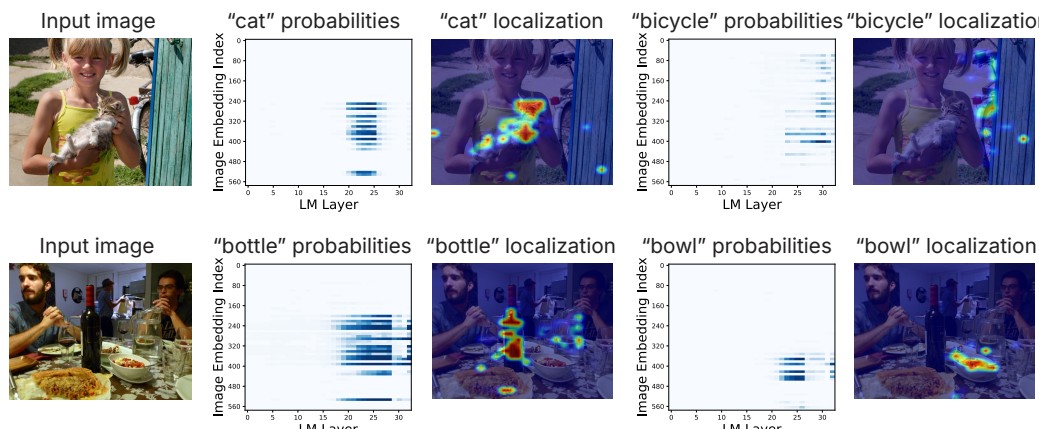

Figure 3: **Localizing objects using internal confidence values**. We find the probabilities of objects through layers of the language model for every image embedding in LLaVA. We use the highest layer probability per image embedding to localize an object within the image.

While the model is not directly trained to map the image representations closer to the text representations of objects within them, we can unembed the image representations in the text vocabulary for localization and find differences in internal confidence for present and hallucinated objects. In Section 5.1, we will use this observation for various applications including hallucination detection and zero-short segmentation.

# 4 ERASING KNOWLEDGE FROM VLMS

Recognizing that image embeddings are directly interpretable (Section 3.2), we edit these embeddings to erase the presence of objects from image captions. We propose a linear editing algorithm that subtracts the text embedding of a target object from all image embeddings. When applied on singular and multiple object removals, we find that it erases hallucinated objects more effectively than correctly detected (CD) objects (i.e. real objects that the model correctly detects).

## 4.1 ERASING OBJECTS FROM IMAGE REPRESENTATIONS

We present an algorithm, PROJECTAWAY (Figure 4), that orthogonalizes image representations with respect to text representations in order to erase objects in image captions, applying it to remove objects one at a time and all at once.

Given an image and an object to remove, we edit the latent representations $h_{l^I}(k_i)$ at a hidden layer $l^I$ across all image embeddings $k_i$. We do not modify any latent representations outside of those belonging to image features. We compute the dot product, $p$, of $h_{l^I}(k_i)$ and the object's text embedding $\vec{t}$, subtracting a weighted $\vec{t}$ from $h_{l^I}(k_i)$ only if the dot product is positive. At $\alpha = 1$, PROJECTAWAY is equivalent to orthogonalizing the image representations with respect to the text representation. To compute text representation $\vec{t}$, we pass the object (e.g. "hot dog") into the VLM's text model and extract $h_{l^T}(t_{-1})$ at hidden layer $l^T$, where $t_{-1}$ is the last token of the object. We use the last token of the object to capture the whole of the object's meaning.

---

**Algorithm 1: PROJECTAWAY**

**Input:** A set of image embeddings $K$, text embedding $\vec{t}$, and weight factor $\alpha$
**Output:** A set of modified image embeddings $K'$ projected away from the text embedding
**Initialization:** $K' \leftarrow \emptyset$
**for** $\vec{k} \in K$ **do**
    $p \leftarrow \vec{k} \cdot \vec{t}$
    **if** $p > 0$ **then**
        $K' \leftarrow K' \cup \{\vec{k} - \alpha \cdot \frac{p}{\|\vec{t}\|_2^2} \cdot \vec{t}\}$
    **else**
        $K' \leftarrow K' \cup \{\vec{k}\}$
    **end if**
**end for**

---

Figure 4: Our editing algorithm erases the presence of an object from image embeddings by orthogonalizing them with respect to the object's text embedding.

| Edit Scope | Model | Individual RR (%) | Mass RR (%) | CD change (%) | $C_i \downarrow$ | $C_s \downarrow$ |
|---|---|---|---|---|---|---|
| No edits | InstructBLIP | - | - | - | 15.0 | 54.1 |
| | LLaVA | - | - | - | 14.6 | 51.1 |
| Hallucinations | InstructBLIP | 83.3 | 74.3 | +0.07 | 8.94 | 33.2 |
| | LLaVA | 86.0 | 72.8 | +0.01 | 11.2 | 35.5 |
| CD | InstructBLIP | 16.2 | 15.0 | -2.2 | 17.3 | 58.3 |
| | LLaVA | 6.9 | 8.3 | -1.6 | 15.2 | 52.4 |

Table 1: **Removing mentioned objects individually & in-mass.** Using PROJECTAWAY, we remove hallucinated objects and observe high hallucination reduction with CHAIR, mass-removal rate (Mass RR), and individual removal rate (Individual RR). We also remove correctly detected (CD) objects but find that they are more resistant to linear editing. Denote CHAIR$_S$ as $C_S$ and CHAIR$_I$ as $C_I$.

### 4.1.1 REMOVING OBJECTS ONE BY ONE

We evaluate the PROJECTAWAY algorithm's effectiveness at erasing individual objects from captions across multiple images and objects.

**Experimental setting.** We apply PROJECTAWAY on 5000 random images from the COCO2014 training set on all mentioned COCO objects (i.e. hallucination and CD) individually and measure the removal rate at which objects no longer appear in the caption. For InstructBLIP, we set $(l^I, l^T, \alpha) = (1, 2, 1.5)$. For LLaVA, we set $(l^I, l^T, \alpha) = (19, 21, 3.5)$. These parameters are fixed irrespective of image and are chosen for their maximal effect (see ablations in Section 4.2). To differentiate hallucinations from CD, we compute CHAIR (Rohrbach et al., 2019), an evaluation criteria that compares model-generated captions to ground-truth human annotations. CHAIR provides two main scores, CHAIR$_I$ and CHAIR$_S$, that quantify hallucinations for instances and sentences, respectively:

$$\text{CHAIR}_S = \frac{|\{\text{captions with hallucinated objects}\}|}{|\{\text{all captions}\}|}, \text{CHAIR}_I = \frac{|\{\text{hallucinated objects}\}|}{|\{\text{all objects mentioned}\}|} \quad (3)$$

**Results.** Table 1 shows that PROJECTAWAY is significantly more effective in erasing individual hallucinated objects at an individual level than CD objects for both InstructBLIP and LLaVA. Along with the insight that hallucinated objects have lower softmax scores (Figure 2), these results suggest that hallucinated objects manifest more weakly in image embeddings and are hence easier to remove than CD objects.

### 4.1.2 MASS-REMOVING OBJECTS

We iteratively apply PROJECTAWAY to a *set* of objects, following the same experimental setup and observing similarly different removal rates for hallucinated objects and CD objects.

**Mass-removing hallucinations.** We mass-remove hallucinations identified with ground truth annotations using PROJECTAWAY. Table 1 shows that editing out all the hallucinations of an image yields a similar removal rate as individually editing out and, importantly, that erasing hallucinated objects together does not interfere with each other. We achieve a hallucination reduction rate of 41.3% for InstructBLIP and 23.3% for LLaVA (see Table 4). Recall count slightly *increases* for both models, indicating that caption accuracy is preserved. This may be because removed hallucinations are replaced with objects the model is more confident in. Qualitative results are in Figure 5.

**Mass removing CD.** We similarly find that applying PROJECTAWAY can successfully remove CD objects when edited all together in Table 1. Furthermore, CHAIR scores minimally change, which indicates that this mass-removal merely erases object presence without eroding caption accuracy. While the removal rate is lower than for hallucinated objects, this insight proves useful when we apply PROJECTAWAY for hallucination reduction in Section 5.2.

### 4.2 ABLATION STUDY: MASS-REMOVING HALLUCINATIONS

We perform ablations on parameters of PROJECTAWAY to improve object removal rate for erasing hallucinations in-mass.

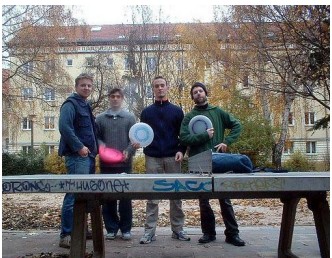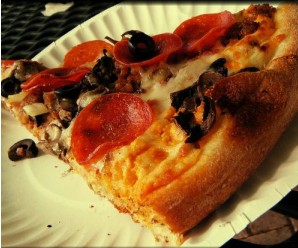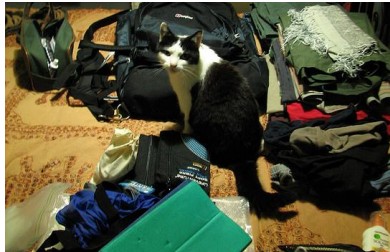

**Before:** In the image, a group of four young **men** are gathered around a picnic table, posing for a photo. They are holding **frisbees** in their hands and seem to be having a good time together. The picnic **table** is located in the middle of the scene, providing a central point for the group's gathering. There are several **chairs** placed around the table, indicating that the group may have been enjoying a meal or socializing while playing with the frisbees. A **car** can be seen parked in the background, possibly belonging to one of the group members.

**After:** A group of four young **men** are gathered around a picnic **table**, posing for a photo while holding frisbees. They seem to be enjoying their time together, possibly playing a friendly game of **frisbee**. The picnic table is located in the middle of the scene, providing a convenient spot for the group to gather and play. In addition to the frisbees, there are several **backpacks** scattered around the area, suggesting that the group may have come prepared for an outdoor adventure.

**Before:** The image features a delicious slice of **pizza** sitting on a white paper plate. The pizza is topped with various ingredients, including pepperoni, olives, and mushrooms, making it a mouth-watering dish. The plate is placed on a **table**, adding to the overall presentation of the pizza. In addition to the pizza slice, there are several other items on the table, including two **cups**, a **fork**, a **knife**, and a **spoon**. These items add to the overall atmosphere of the scene, creating a cozy and inviting setting for enjoying the pizza.

**After:** The image features a delicious slice of **pizza** on a paper plate, placed on a **dining table**. The pizza is topped with various ingredients, including pepperoni, olives, and mushrooms, making it a mouth-watering meal. The pizza slice dominates the scene, taking up most of the space on the paper plate. The plate is placed on top of a dining table, which can be seen in the background.
.

**Before:** The image features a black and white **cat** sitting in the middle of a cluttered living room floor. The cat is surrounded by various items such as clothes, bags, **books**, and a **backpack**. There are two **chairs** placed near the cat, one on the left side and the other on the right side of the room. In addition to the cat, there are two people visible in the scene. One **person** is standing on the left side of the room, while the other person is located on the right side of the room. Both individuals seem to be engrossed in their own activities, possibly unaware of the cat's presence.

**After:** The image depicts a black and white **cat** sitting in the middle of a cluttered room. The cat is surrounded by a variety of items, including **suitcases**, **backpacks**, clothes, and shoes. There are at least three suitcases scattered around the room, with one located closer to the cat and the other two further away. A backpack can be seen on the left side of the room, and a pair of shoes can be spotted on the right side. In addition to these items, there are several clothes spread out on the floor, including a shirt, a jacket, and a pair of pants. The cluttered environment suggests that the room may have been recently used for packing or preparing for a trip.

Figure 5: **Qualitative results for mass object removal.** We present example images and their captions after mass-removing hallucinations (red) with PROJECTAWAY., which can effectively remove hallucinations while preserving, even increasing, correctly detected objects (green).

**Experimental setting.** We ablate the three parameters of PROJECTAWAY: layer $l^I$ to edit at, layer $l^T$ to retrieve the text representation, and weight factor $\alpha$. At $l^T = -1$, we average together the object's constituent token embeddings. At $l^I = -1$, we edit the image embeddings directly inputted to the text model. We evaluate across 500 training samples from COCO 2014 that have at least one hallucination.

**Hidden layers.** Figure 6 shows hallucination reduction rate on LLaVA from mass-removing hallucinations on every combination of $l^I$ and $l^T$ (each from -1 to 31). As a core concern is that editing erodes caption accuracy, we gray out any combination that reduces CD objects. For InstructBLIP (see Figure 10), the best parameters ($l^I = 1, l^T = 2$) reduces hallucinations by 38.5%. For LLaVA, our best parameters ($l^I = 19, l^T = 21$) reduce hallucinations by 25.7%, and the middle layers are the best to edit and extract latent text embeddings from. Our results also provide a wide range of reasonable parameter alternatives to use if this reduction rate does not generalize beyond our samples.

**Weight factor.** Using the best-reduced hidden layers, we ablate the weight factor $\alpha$ for PROJECT-AWAY across the same 500 randomly selected COCO images. Figure 7 shows that as $\alpha$ increases, hallucinations are removed at a higher rate, and the overall hallucination count drops significantly. At high $\alpha$, we observe through anecdotal examples that captions become nonsensical, as quantitatively shown by the complete loss of both correctly detected and hallucinated objects from the caption. Therefore, as a pre-caution, we only select weight factors that do not reduce CD objects when we apply PROJECTAWAY to erase hallucinated objects.

# 5 APPLICATIONS

## 5.1 HALLUCINATION DETECTION

When extracting knowledge from VLMs in Section 3.2, we found that applying logit lens on in-context image representations exhibit useful information about visual objects present in the image. Using these observations, we present an approach for object presence classification that only relies on the VLMs own parameters. We utilize the internal confidence $c_o$ value to classify object presence,

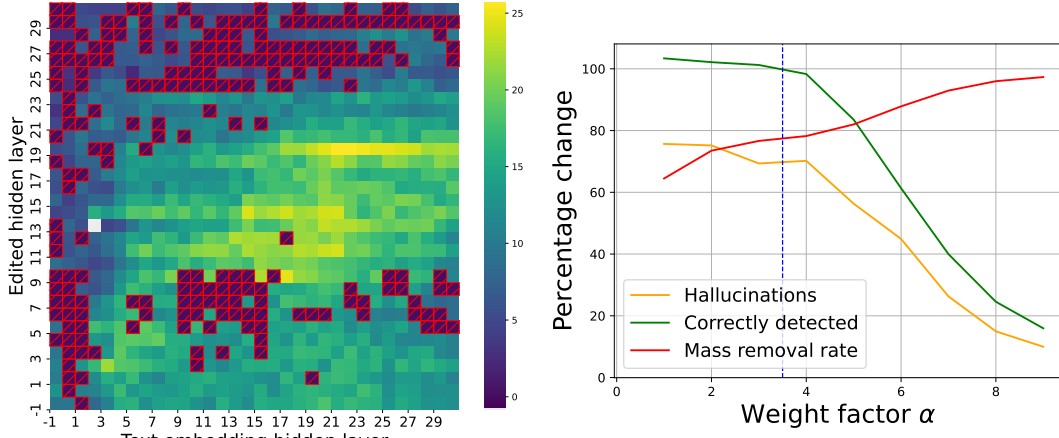

Figure 6: **Hidden layer ablations** for LLaVA. We track hallucination reduction (%) across different layers to edit at and extract latent embeddings for the text embedding, crossing out (red) parameters from consideration where there is a decrease in correctly detected objects.

Figure 7: **Weight ablations** for LLaVA. We vary the weight factor $\alpha$ and measure changes in correctly detected objects, removal rate, and hallucination reduction. We observe a decline in hallucinations as weight grows and mark a weight where there is no loss in correctly detected objects.

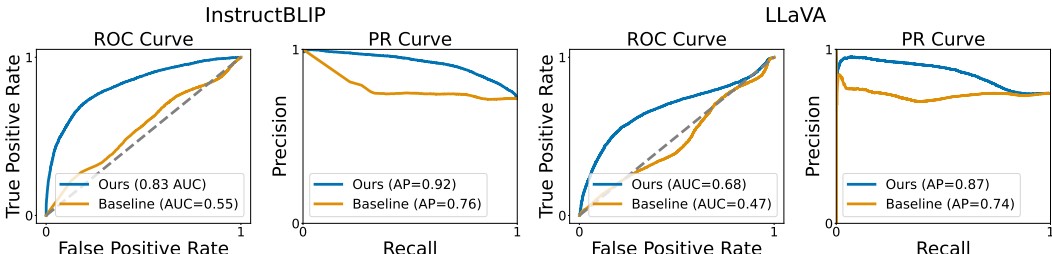

Figure 8: **Object Presence Classification Curves for InstructBLIP and LLaVA.** We show the Precision-Recall and ROC curves of our confidence measure for present object-hallucination classification on the COCO training subset. Classifying object presence with the internal confidence outperforms the baseline, indicating that the language model's image representations know which objects are hallucinations and which are truly present.

since the internal confidence for objects that are not present in the image, or hallucinated, are lower within the image representations.

**Experimental setting.** We evaluate the strength of the internal confidence $c_o$ as an indicator of object presence. We sample 5000 images from the MSCOCO training set, using the image captioning objective to caption methods with both InstructBLIP and LLaVA. We use the $c_o$ for present objects and hallucinations within the captions generated by each VLM. We assess how well the internal confidence aligns with the ground truth labels of object presence, where a negative sample is a hallucination and a positive sample is a present object.

**Baseline.** As a baseline, we use the maximum output probability of the object's tokens. This is the confidence of the model prediction. Previous works such as Zhou et al. (2024) have found that hallucinations occur more frequently on objects characterized by high uncertainty during generation.

**Results.** We present quantitative results in Figure 8 and Table 5. We show qualitative results for LLaVA (Figure 14) and InstructBLIP (Figure 15) in the Appendix. We find that utilizing internal confidence to classify object hallucinations provides a 47.17% improvement in mAP in InstructBLIP and 22.45% in LLaVA. Furthermore, the ROC AUC improves over the baseline by 50.10% in InstructBLIP and 44.68% in LLaVA, indicating stronger object presence classification.

| Model | Method | CHAIR$_i$ ↓ | CHAIR$_s$ ↓ | Hallucinated Objects ↓ |
|---|---|---|---|---|
| InstructBLIP | Greedy | 57.0 | 23.3 | 512 |
| | Nucleus | 58.0 | 24.0 | 508 |
| | Beam Search | 53.4 | 14.6 | 564 |
| | OPERA | 45.6 | 13.9 | 472 |
| | Ours | **43.8** | **12.5** | **419** |
| LLaVA | Greedy | 49.2 | 14.2 | 532 |
| | Nucleus | 55.8 | 17.1 | 618 |
| | Beam Search | 52.4 | 15.0 | 583 |
| | OPERA | 44.8 | 12.8 | 462 |
| | Ours | **42.0** | **12.2** | **444** |

Table 2: **Hallucination intervention performance.** We mass-remove hallucinations detected by the method in Section 5.1 and outperform other baselines. We observe a considerable drop in the raw count of hallucinated objects.

## 5.2 HALLUCINATION REMOVAL

We use the mass editing technique to remove hallucinations detected by the prior method. Section 4.1.2 successfully removes a significant portion of hallucinations but presupposes a knowledge of what the hallucinations are. We threshold on the internal confidence of each object to identify hallucinations and mass-remove them using PROJECTAWAY. Our chosen threshold prioritizes precision over recall (i.e. we allow classification of some CD objects as hallucinations) because CD objects are less affected by the removal method, as shown in Section 4.1.2.

**Experimental setting.** We threshold hallucinations as $c_o < 0.2$ for InstructBLIP and $c_o < 0.1$ for LLaVA. Based on prior ablations (Section 4.2), we select $(l^I = 1, l^T = 2, \alpha = 1.5)$ for InstructBLIP and $(l^I = 19, l^T = 21, \alpha = 3.5)$ for LLaVA. Our prompt is "Please describe this image in detail."

**Baselines.** Since our method intervenes during the decoder step, we compare our method with 3 standard decoding algorithms. Greedy decoding predicts the next token based on the highest logit probability. Beam search maintains a tree of beams and selects the best beam at generation end. Nucleus sampling selects the next token from a set of high probability tokens whose cumulative probability reaches a threshold $p$. We also evaluate against OPERA (Huang et al., 2024), which mitigates hallucinations by adding an overtrust penalty during decoder generation. We set $p = 0.9$ for nucleus sampling. We use beam search in our method and unify $N_{\text{beam}} = 5$ for the baseline.

**Results.** We apply these parameters to 500 COCO images from the Karpathy validation set. We provide qualitative results in Figure 17 and Figure 16. Quantitative results in Table 2 show that we outperform our baselines and reduce hallucinations by 25.7% on InstructBLIP and 23.8% on LLaVA compared to beam search. Our approach achieves a similar hallucination reduction rate as Section 4.1.2, despite not precisely differentiating hallucinations and some CD objects being incorrectly edited out. Notably, our method relies on no training or external models, effectively offering a "free lunch." We find similar performance on additional models (Appendix A.5) and attribute hallucinations (Appendix A.7).

## 5.3 ZERO-SHOT SEGMENTATION

Building upon our findings in Section 3.2, we utilize the internal confidence per image feature for zero-shot image segmentation. This application leverages the spatial information encoded in the image representations and demonstrates how VLMs internally represent and localize objects within images.

**Method.** Our approach leverages the spatial correspondence between image patches and their associated image embeddings. We use LLaVA to generate the name of the class in the image and we focus on the internal confidence of that class per image patch. We take the mean internal confidence for tokens comprising a class word. We resize the set of $24 \times 24$ internal confidence values per image patch back into a fixed image size of $336 \times 366$ pixels. We then apply a threshold to these confidence values to binarize them into a foreground/background segmentation for the object in the image.

| Model | Method | Pixel Acc. ↑ | mIoU ↑ | mAP ↑ |
|---|---|---|---|---|
| raw attention (CLIP) | Image Encoder | 69.81 | 45.19 | 77.30 |
| TextSpan (Gandelsman et al., 2024b) | Image Encoder | 75.57 | 53.60 | **80.22** |
| raw attention (VLM) | VLM | 67.28 | 39.27 | 73.96 |
| Ours | VLM | **76.16** | **54.26** | 79.90 |

Table 3: **Segmentation Performance on ImageNet-segmentation.** Localizing objects using their probabilities within the image representations results in more accurate zero-shot segmentation than previous methods relying on vision encoders and VLMs.

**Baseline.** As a baseline, we extract the attention values of generated tokens with the image embeddings from LLaVA. We also compare to the segmentation method introduced by Gandelsman et al. (2024b), which utilizes the attention heads of the image encoder without the additional VLM processing, using the same image encoder (CLIP-ViT-L/14 at 336px).

**Results.** We evaluate our method on the Imagenet validation set. Qualitative results are shown in Figure 9 and quantitative comparisons with the baselines in Section 5.3. We improve mAP by 8.03% over using the VLMs raw attention values and provide better and/or comparable performance to other state-of-the-art methods that utilize just the image encoder. While the VLM is not directly trained for segmentation, our technique reveals that it still encodes significant *spatial* information about objects within its intermediate image representations.

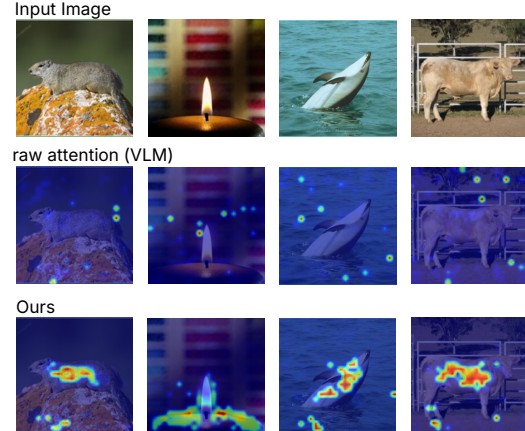

Figure 9: **Zero-shot segmentation.** Warmer areas indicate higher internal confidence for the class at that image patch. We binarize these values with a threshold to generate segmentations.

# 6 DISCUSSION AND LIMITATIONS

We interpreted VLMs' image representations through the language model layers and discovered that linear editing of these representations can selectively remove object information via a simple orthogonalization. Our findings enabled hallucination reduction and improved zero-shot segmentation. We present two limitations of our work and conclude with future directions.

**Multi-token objects.** Our method simplifies the use of object words that may be composed of multiple tokens, such as by taking the max internal confidence over object tokens or utilizing the average token embedding for editing. This can introduce noise to the internal confidence if certain tokens are common in multiple different words and lead to an approximation of the object's latent representations when editing.

**Fine-grained edits.** The editing approach may struggle with highly abstract or longer sentences that involve attributes or interactions of objects. Removing a full sentence, for example, is not something we assessed in this paper, since our focus is on the removal of individual objects.

**Future work.** While our focus was on interpreting objects and object hallucinations in VLMs, we believe that our approach can be extended to other key elements of visual scenes, such as people, attributes, and actions. We also focused on object removal, but we believe that editing can also be extended to inject objects into a caption (by adding instead of subtracting the text embedding). We hope to explore the applications of our approach in other multimodal architectures. Our insights may help design better VLMs that are more robust to hallucinations and have improved spatial understanding. We plan to explore these directions in our future work.

## 6.1 ACKNOWLEDGMENTS

We thank Kayo Yin for her comments and feedback on our paper. YG is supported by the Google Fellowship. Authors, as part of their affiliation with UC Berkeley, were supported in part by the the Berkeley Artificial Intelligence Research (BAIR) commons program.

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

| Edit Scope | Model | Hallucinations | CD |
|---|---|---|---|
| No edits | InstructBLIP | 4545 | 14178 |
| | LLaVA | 4372 | 15053 |
| Hallucinations | InstructBLIP | 2672 | 14189 |
| | LLaVA | 3348 | 15061 |
| CD | InstructBLIP | 5078 | 13864 |
| | LLaVA | 4583 | 14826 |

Table 4: **Supplemental metrics for Table 1.** We measure unique hallucinated and correctly detected (CD) objects.

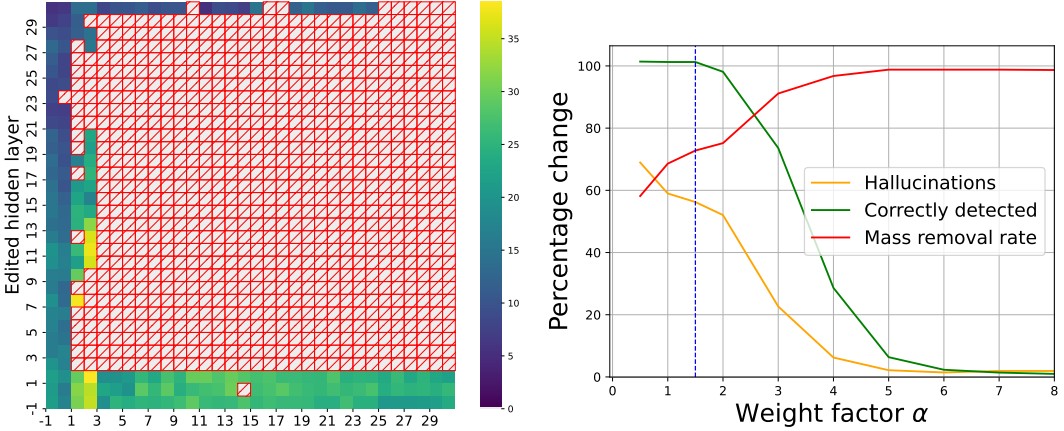

Figure 10: **Hidden layer ablations for InstructBLIP**. We track hallucination reduction (%) across different layers to edit at and extract latent embeddings for the text embedding, crossing out (red) parameters from consideration where there is a decrease in correctly detected objects.

Figure 11: **Weight ablations for InstructBLIP.** We vary the weight factor $\alpha$ and measure changes in correctly detected objects, object removal rate, and hallucination reduction. We observe a decline in hallucinations as weight increases and mark a weight where there is no loss in correctly detected objects.

# A APPENDIX

## A.1 MASS-REMOVING OBJECTS

We mass-remove mentioned objects (hallucinations and correctly detected) with PROJECTAWAY and tally up the total number of unique hallucinated and CD objects in Table 4.

## A.2 ABLATIONS FOR INSTRUCTBLIP

We show hidden layer and weight ablations for mass-removing hallucinations in InstructBLIP referenced in Section 4.2. The hidden layer ablations indicate that most of the parameter space is too sensitive to edit and leads to losses in correctly detected objects. We find that smaller $l^T$ and $l^I$ parameters are the most effective for reducing hallucinations. Our best parameters ($l^I = 1, l^T = 2$) reduce hallucinations by 38.5%. It is not fully understood why the majority of the parameter search space is invalid in comparison with LLaVA in Figure 6. It is possible that the fine-tuning step in LLaVA semantically aligns hidden image representations with text embeddings more than InstructBLIP, allowing linear edits to have the precise, intended effect.

### A.3 HALLUCINATION DETECTION

We show quantitative comparisons from our hallucination detection approach using internal confidence (Section 5.1) to the baseline in Table 5. We also show qualitative examples for LLaVA in Figure 14 and for InstructBLIP in Figure 15. These samples exhibit model-generated captions, parsed objects, and whether they are classified as hallucinated or correctly detected based on their internal confidence score.

### A.4 HALLUCINATION REDUCTION

We exhibit sample results from our hallucination reduction approach (Section 5.2), which linearly removes text representations of hallucinations from image representations, in Figure 17 for Instruct-BLIP and Figure 16 for LLaVA. We show the image caption before and after our linear editing method, removing objects detected as hallucinations.

### A.5 QUANTITATIVE EVALUATIONS ON MORE ADVANCED MODELS

We evaluate our approach on two additional models, LLaVA-NeXT 7B (Liu et al., 2024b) and Cambrian-1 8B (Tong et al., 2024) with Llama 3. We threshold hallucinations as $c_o < 0.4$ for LLaVA-NeXT and $c_o < 0.3$ for Cambrian-1. Based on qualitative examples and referencing optimal parameters from other models in Section 4.2, we select ($l^I = 24, l^T = 22, \alpha = 2$) for both models. We show quantitative results for hallucination detection in Table 6 and for hallucination intervention in Table 7. With our method, we observe a 27.73% improvement in $CHAIR_S$ with LLaVA-NeXT and a 28.86% improvement with Cambrian-1, demonstrating consistency with our findings on the LLaVA and InstructBLIP models.

### A.6 OBJECT LOCALIZATION

We show qualitative examples for localization with internal confidence for specific image representations, specifically for the LLaVA model, in Figure 18.

### A.7 ATTRIBUTE HALLUCINATIONS

Our analysis in this paper centered on object hallucinations because automated tooling and benchmarks for attribute (ex. shape, color, number) hallucinations are relatively sparse. However, we demonstrate the applicability of our editing technique on attribute hallucinations with qualitative examples filtered from the VQA 2.0 challenge in Figure 12. We reuse the editing hyperparameters for InstructBLIP ($l_I = 1, l_T = 2, \alpha = 1.5$) and only edit attributes with $c_o < 0.05$.

### A.8 ZERO-SHOT CLASSIFICATION

We evaluate the strength of internal confidence derived from the logit lens on image representations for classification of the COCO class within patches of the image. We use the COCO ground truth segmentations to find ground truth classes for image patches. We determine the accuracy of the rankings found from logit lens internal confidence scores to predict the class per patch and present our results in Table 8. We find that these values highly vary across classes, which we hypothesize is because certain classes such as "person" are represented with more specific tokens such as "doctor", "skier", "girl", etc. resulting in lower internal confidence for the tokens in "person" while other objects like "toothbrush", "banana", and "broccoli" are described in the same word as the COCO class.

### A.9 QUALITATIVE EXAMPLES BEYOND COCO 2014

We focus on COCO 2014 in our analyses because CHAIR, our main evaluation criteria, is tied with the dataset and can automatically categorize objects of interest in image captions. While COCO 2014 is a diverse set of images, we provide qualitative examples of hallucination reduction (see Section 5.2) on images from LLaVA-Bench (Liu et al., 2024b), a collection of 24 images of varying subjects. The examples in Figure 13 using InstructBLIP align with the strong hallucination reduction observed with COCO 2014.

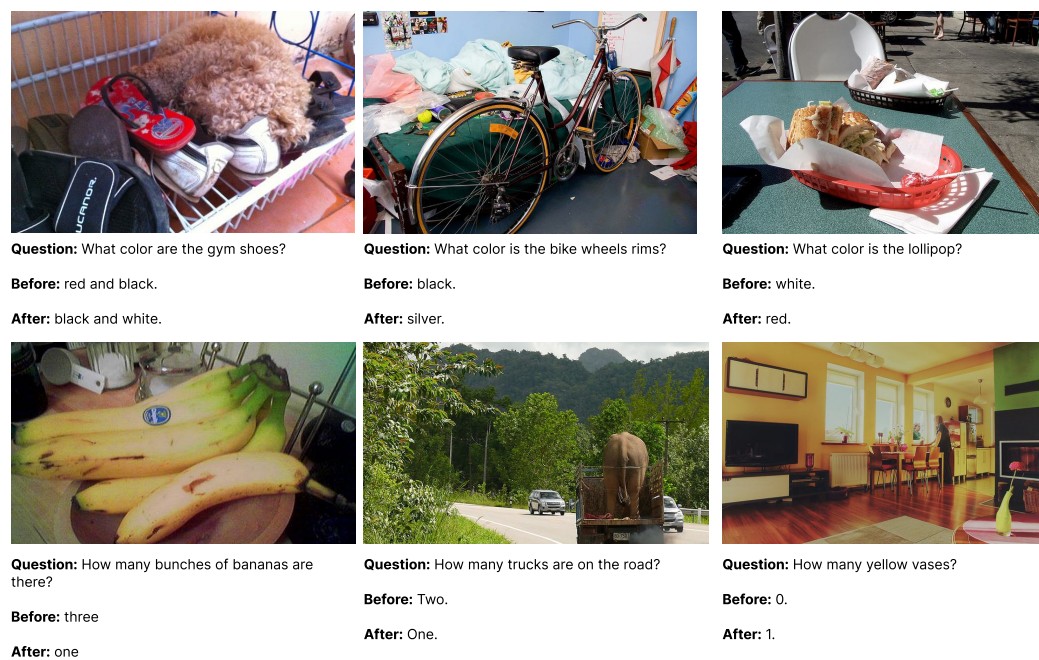

Figure 12: **Qualitative results for attribute hallucinations using InstructBLIP**. We filter the VQA dataset for color and object number inaccuracies and correct answers with low confidence scores ($c_o < 0.05$) using PROJECTAWAY. We reuse the same hyperparameters previously chosen for InstructBLIP ($l^I = 1, l^T = 2, \alpha = 1.5$).

| Method | InstructBLIP | | LLaVA | |
|---|---|---|---|---|
| | mAP ↑ | ROC AUC ↑ | mAP ↑ | ROC AUC ↑ |
| Baseline | 0.53 | 0.55 | 0.49 | 0.47 |
| Ours | 0.78 | 0.83 | 0.60 | 0.68 |

Table 5: **Object presence classification performance.** We use internal confidence $c_o$ as a confidence score to classify whether the object is present in the image. We evaluate the mAP and ROC AUC of our classification method against the baseline for both the InstructBLIP and LLaVA models over a subset of 5000 COCO images.

| Method | LLaVA-NeXT | | Cambrian-1 | |
|---|---|---|---|---|
| | mAP ↑ | ROC AUC ↑ | mAP ↑ | ROC AUC ↑ |
| Baseline | 0.93 | 0.66 | 0.94 | 0.73 |
| Ours | 0.95 | 0.75 | 0.97 | 0.83 |

Table 6: **Object presence classification on more models.** We classify whether the object is present in the image using internal confidence for LLaVA-NeXT and Cambrian-1 over a subset of 500 COCO images.

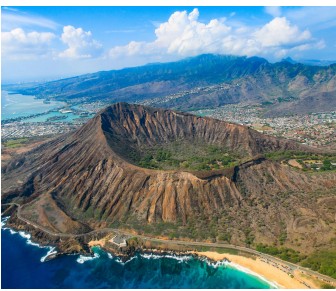 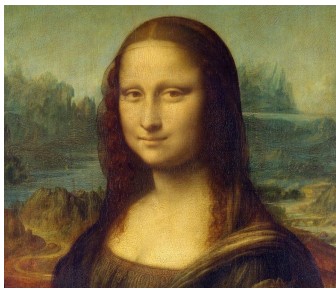 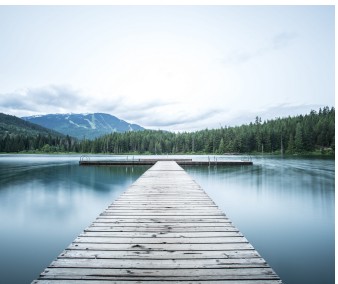

**Before:** The image captures a breathtaking aerial view of a scenic coastline with a **mountainous** landscape in the background. The **ocean** is visible in the foreground, stretching from the left side of the image to the right. There are several **boats** scattered throughout the water, some closer to the shoreline and others further out. In addition to the boats, there are several **people** visible in the scene. Two people can be seen on the left side of the image, one closer to the water and the other slightly further away. On the right side of the image, there are two more people, both closer to the water's edge.

**After:** The image depicts an aerial view of a scenic coastal area with a large **mountain** in the background. The mountain is situated on the left side of the image, while the **ocean** can be seen on the right side. There are several **cars** visible in the scene, likely parked or driving along the coastal road. Some of the cars are closer to the mountain, while others are further away from it. In addition to the cars, there are several **trees** scattered throughout the landscape, adding to the natural beauty of the area.

**Before:** The image features a **painting** of a beautiful **woman** with long, flowing hair. She is depicted wearing a black dress and has a serene expression on her face. The painting captures the essence of Leonardo da Vinci's Mona Lisa, one of the most famous paintings in the world. In the painting, the woman's face takes up a significant portion of the canvas, making her the focal point of the artwork. Additionally, there are several **trees** visible in the background of the painting, adding depth and dimension to the scene.

**After:** The image depicts a **painting** of a beautiful **woman** with long, flowing hair. She is wearing a black **dress** and has a serene expression on her face. The painting captures the essence of the famous painting "Mona Lisa" by Leonardo da Vinci, which is considered one of the most famous paintings in the world. The painting is well-known for its enigmatic smile and intricate brushstrokes, making it a timeless masterpiece.

**Before:** The image depicts a long wooden **pier** extending into a serene **lake**, surrounded by **trees** and **mountains** in the background. The pier stretches out into the water, inviting visitors to explore the tranquil surroundings. A **boat** can be seen docked at the end of the pier, adding to the peaceful atmosphere of the scene. In addition to the boat, there are several smaller boats scattered around the lake, creating a sense of activity and enjoyment for those spending time on the water. Overall, the image captures a relaxing and picturesque setting, perfect for unwinding and taking in the natural beauty of the lake and its surroundings.

**After:** The image depicts a long wooden pier extending into a serene **lake**, surrounded by **trees** and **mountains** in the background. The **pier** is situated in the middle of the lake, creating a peaceful and tranquil atmosphere. There are several trees visible in the surrounding area, adding to the natural beauty of the scene. The pier seems to be a popular spot for fishing or simply enjoying the serene surroundings.

Figure 13: **Qualitative results on images from LLaVA-Bench**. We randomly select images from the benchmark and use InstructBLIP to detect and edit out hallucinations. Our hyperparameter selection is the same as in Section 4.1.1 ($l^I = 1, l^T = 2, \alpha = 1.5$).

| Model | Method | CHAIR$_i$ ↓ | CHAIR$_s$ ↓ | Recall (%) ↑ |
|---|---|---|---|---|
| LLaVA-NeXT | Beam Search | 6.8 | 23.8 | 63.12 |
| | Ours | **5.52** | **17.2** | 63.12 |
| Cambrian-1 | Beam Search | 3.27 | 9.2 | 53.28 |
| | Ours | **2.7** | **6.6** | 53.28 |

Table 7: **Hallucination intervention performance on more models.** We mass-remove hallucinations detected by the method in Section 5.1 on two more models, LLaVA-NeXT and Cambrian-1, on the same subset of 500 COCO images as used in Table 2. We observe consistent improvement over the baseline while maintaining recall of objects present in the image.

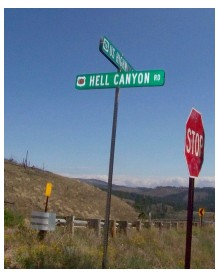

**Internal Confidences**
stop sign 99.27%
car 2.54%
truck 2.34%
pedestrian 0.00%

**Baseline Caption**
The image depicts a street corner with a **stop sign** prominently displayed on the right side of the scene. Next to the stop sign, there is a street sign that reads "Hell Canyon Rd." The street sign is positioned above the stop sign, making it clearly visible for drivers and **pedestrians**. In the background, there is a **car** parked on the left side of the scene, and a **truck** can be seen further back on the right side.

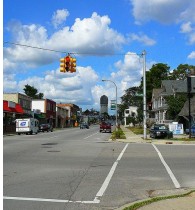

**Internal Confidences**
traffic light 97.61%
truck 90.28%
car 47.20%
fire hydrant 4.02%
person 0.44%

**Baseline Caption**
The image depicts a quiet city street on a sunny day, with a few **cars** and **trucks** parked along the side of the road. There are multiple **traffic lights** at various points along the street, ensuring safe passage for vehicles and pedestrians.

In addition to the cars and trucks, there are a few **pedestrians** walking along the sidewalk. One **person** can be seen near the center of the scene, while two others are closer to the right side of the image. A **fire hydrant** is also visible on the right side of the street. Overall, the scene conveys a calm and peaceful atmosphere in the city.

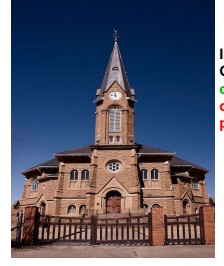

**Internal Confidences**
clock 98.04%
car 7.20%
person 2.10%

**Baseline Caption**
The image features a large brick church with a tall **clock** tower. The clock is situated on the side of the tower, making it a prominent feature of the building. The church has a steeple and a cross on top of the tower, adding to its architectural charm.
There are two **people** visible in the scene, one standing closer to the left side of the church and the other closer to the right side. Additionally, there is a **car** parked in front of the church, likely belonging to one of the visitors.

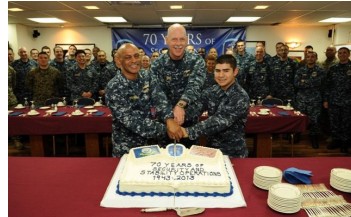

**Internal Confidences**
cake 99.99%          man 96.33%
table 88.02%         chair 57.04%
cup 6.85%            tv 0.686%

**Baseline Caption**
The image depicts a group of **men** in military uniforms gathered around a dining table, celebrating their 70th anniversary. They are shaking hands and posing for a picture in front of a large sheet **cake**. The cake is placed on the table, taking up a significant portion of the space.
There are multiple **chairs** surrounding the **dining table**, and several **cups** can be seen on the table as well. In the background, a **TV** is mounted on the wall, adding to the atmosphere of the celebration.

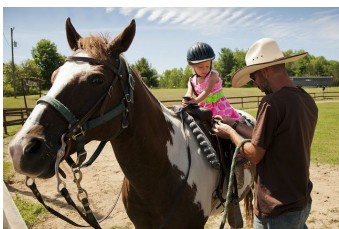

**Internal Confidences**
horse 96.33%         man 60.63%
girl 29.65%          bench 9.58%
dining table 2.43%

**Baseline Caption**
The image features a **man** and a young **girl** sitting on the back of a brown and white **horse**. The man is holding the reins of the horse, ensuring the girl's safety while she enjoys the ride. The girl is wearing a pink dress, adding a touch of color to the scene.
In the background, there is a **bench** and a **dining table**, suggesting that the location might be a park or an outdoor gathering area.

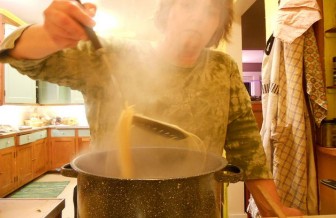

**Internal Confidences**
oven 98.22%          bottle 97.34%
woman 83.76%         stove 71.44%
spoon 64.54%         refrigerator 6.61%
microwave 4.93%      bowl 3.02%

**Baseline Caption**
The image features a **woman** standing in a kitchen, cooking a meal. She is wearing a green shirt and is focused on stirring a large pot on the **stove**. The pot appears to be filled with food, and the woman is using a **spoon** to mix the ingredients.
The kitchen is well-equipped with various appliances and utensils. There is a **refrigerator** on the left side of the room, and an **oven** can be seen on the right side. A **microwave** is placed above the oven, and a **bowl** is located on the countertop near the center of the kitchen. Additionally, there are two **bottles**, one on the left side and the other on the right side of the kitchen, possibly containing cooking ingredients or beverages.

Figure 14: **LLaVA Object Presence Classification**. Sample image captions from LLaVA and the internal confidence scores for objects in the caption used for classification as correctly detected objects or hallucinations.

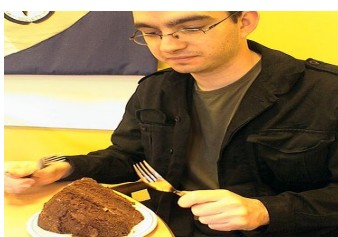 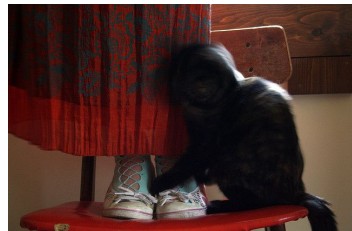 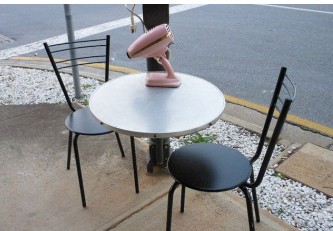

**Internal Confidences**
cake 99.76%    fork 97.46%
table 94.19%    man 25.49%
knife 1.79%    spoon 0.44%
bowl 0.4%

**Baseline Caption**
The image features a **man** sitting at a dining **table** with a plate of chocolate **cake** in front of him. He is holding a **fork** and appears to be about to take a bite from the cake. There are several other utensils on the table, including **knives**, forks, and **spoons**, suggesting that the man might be sharing the cake with someone else. In the background, there is a **bowl** placed on the table, possibly containing additional food or drinks. Overall, the scene depicts a man enjoying a delicious slice of chocolate cake.

**Internal Confidences**
cat 95.21%    chair 88.33%
table 4.16%    bowl 0.32%
cup 0.13%

**Baseline Caption**
The image features a black **cat** sitting on a red **chair** next to a pair of white shoes. The cat is interacting with the shoes, possibly trying to play with them or investigate them. There are several other objects in the room, such as a **bowl** and a **cup** placed on a **table** near the cat's position. Additionally, a window can be seen in the background, adding to the overall atmosphere of the scene.

**Internal Confidences**
table 98.78%    chair 73.14%
umbrella 4.15%    person 0.47%
car 0.18%

**Baseline Caption**
The image features a small **table** and two **chairs** placed outside on a sidewalk. The table has a vintage hair dryer sitting on top of it, adding a unique touch to the scene. The chairs are positioned close to the table, creating a cozy seating area for two **people**. There is also an **umbrella** placed near the table, providing additional protection from the sun or rain. A street can be seen in the background, with **cars** passing by occasionally.

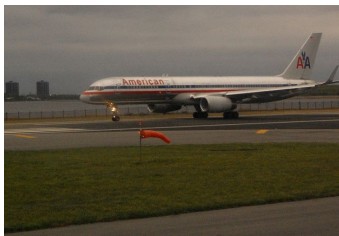 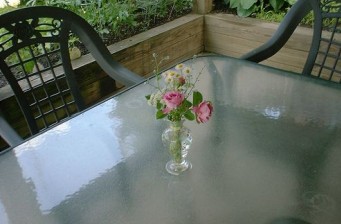 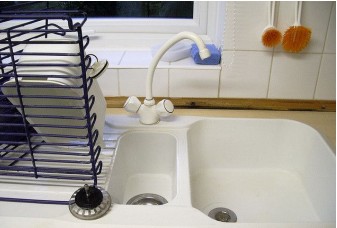

**Internal Confidences**
airplane 56.59%    plane 19.46%
person 0.53%    fire hydrant
0.23%

**Baseline Caption**
The image depicts an American Airlines **airplane** parked on a runway next to a body of water, possibly a lake or a river. The **plane** appears to be preparing for takeoff or landing. There are several **people** visible in the scene, some standing near the airplane, while others are positioned further away from it. In total, there are at least 10 people present in the scene. Additionally, there is a **fire hydrant** located close to the airplane, likely for emergency purposes.

**Internal Confidences**
table 99.85%    vase 99.56%
chair 83.84%    bench 3.69%
person 0.22%

**Baseline Caption**
The image features an outdoor patio with a glass-topped dining **table** surrounded by **chairs**. On the table, there is a **vase** filled with pink flowers, adding a touch of beauty and elegance to the space. The vase is placed in the center of the table, making it the focal point of the area. There are several chairs surrounding the table, providing ample seating for **people** to enjoy their meals or relax in the sunshine. A **bench** can also be seen on the left side of the patio, offering additional seating options. Overall, the scene radiates a warm and inviting atmosphere, perfect for outdoor gatherings and relaxation.

**Internal Confidences**
sink 97.07%    spoon 5.19%
knife 0.3%    fork 0.07%

**Baseline Caption**
The image depicts a kitchen sink area with two **sinks**, one on the left side and the other on the right side. The sink on the left is closer to the edge of the countertop, while the sink on the right is positioned further away from the edge. In addition to the sinks, there is a dishwasher located in the middle of the kitchen area. There are several utensils scattered throughout the scene, including a **spoon**, a **fork**, and a **knife**, placed near the sinks and the dishwasher.

Figure 15: **InstructBLIP Object Presence Classification**.

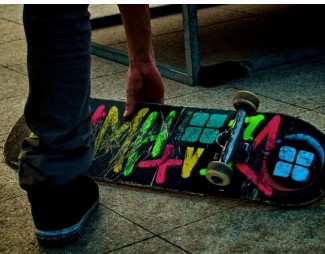

**Before:** The image depicts a **person** wearing a black shirt and gray pants, standing next to a **skateboard**. The skateboard is placed on the ground, with the person's hand resting on it. The person appears to be in the process of picking up the skateboard or preparing to ride it.

In the background, there is a **chair** and a **dining table**, suggesting that the scene might be taking place in an outdoor or semi-outdoor setting.

**After:** In the image, a **person** is standing on a sidewalk, holding a colorful skateboard in their hand. The **skateboard** features a vibrant design, making it an eye-catching sight. The person appears to be either preparing to skate or taking a break from their skateboarding session. The scene captures the essence of skateboarding culture and the enjoyment it brings to those who participate in it.

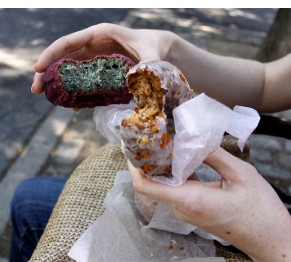

**Before:** The image shows a **person** sitting on a **bench**, holding a donut in their hand. The donut appears to be half-eaten, with a bite taken out of it. The person seems to be enjoying their snack while sitting on the bench.

In addition to the person and the donut, there is a **handbag** placed next to the person on the bench, and a **backpack** can be seen in the background.

**After:** In the image, a **person** is sitting on a **bench** and holding a pastry in their hand. The pastry appears to be a doughnut, and the person is in the process of eating it. Another doughnut can be seen on the bench, close to the person's hand. A **chair** is also visible in the scene, placed near the bench where the person is sitting.

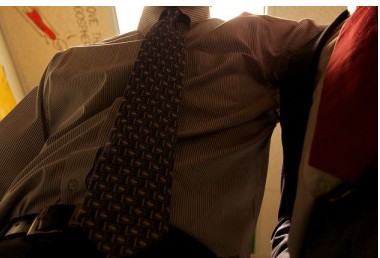

**Before:** The image features a **man** wearing a dress shirt and tie, standing in a room. He is wearing a striped **tie**, and the shirt appears to be white. The man appears to be looking down at his tie, possibly adjusting it or checking its appearance.

In the room, there is a **couch** located towards the right side of the image, and a **chair** can be seen in the background on the left side. The man is positioned in the center of the scene, drawing attention to his attire.

**After:** The image depicts a **man** wearing a dress shirt and **tie**, standing in a room. The shirt is white, and the tie is blue. The man is wearing a belt, which is visible around his waist. The room appears to be dimly lit, creating a somewhat mysterious atmosphere.\n\nIn addition to the man, there are two other people in the room, one on the left side and the other on the right side of the image. However, they are not the main focus of the scene, as the man in the dress shirt and tie is the central figure.

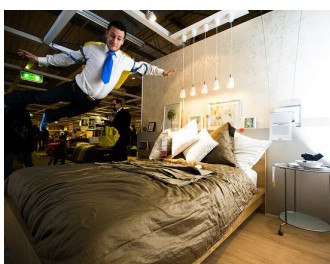

**Before:** The image captures a **man** in mid-air, leaping over a **bed** in a bedroom. He is wearing a **tie** and appears to be in the middle of an energetic jump. There are several other people in the room, with one person standing near the left side of the bed and two others closer to the right side. The bedroom is furnished with a **couch**, a **chair**, and a **dining table**. A **potted plant** can be seen on the right side of the room, adding a touch of greenery to the space. A **clock** is mounted on the wall, and a **cup** is placed on the dining table.

**After:** The image depicts a **man** in a white shirt and a blue **tie** jumping in the air above a bed in a bedroom. He appears to be in the process of diving onto the bed. There are several other people in the room, with some standing near the bed and others scattered around the area. The bedroom is furnished with a **dining table**, a **chair**, and a **potted plant**. There is also a **cup** placed on the dining table, and a **handbag** can be seen near one of the people in the room. Overall, the scene captures a lively and energetic atmosphere in the bedroom.

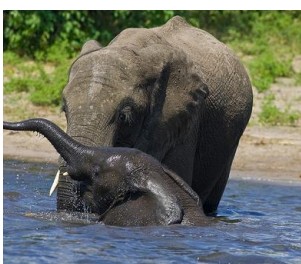

**Before:** The image captures a heartwarming scene of two **elephants**, an adult and a baby, playing together in a body of water. The **adult** elephant is partially submerged in the water, while the baby elephant is swimming nearby. They appear to be enjoying their time in the water, splashing and interacting with each other.

In addition to the two elephants, there are a few other objects in the scene. A **bench** can be seen on the left side of the image, and a **potted plant** is located near the top left corner.

**After:** The image depicts two **elephants** in a body of water, possibly a river or a lake. One of the elephants appears to be an **adult**, while the other is a baby elephant. The baby elephant is swimming in the water, while the adult elephant is standing nearby. They seem to be enjoying their time together in the water.

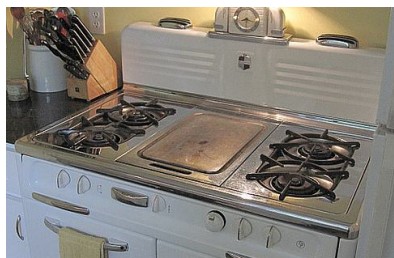

**Before:** The image depicts a clean and well-maintained kitchen with a white **stove** top oven. The **oven** has two burners, one on the left side and the other on the right side. Above the oven, there is a light that illuminates the cooking area. The kitchen is equipped with various utensils, including multiple **knives** and a **spoon**. The knives are placed in a knife block, while the spoon can be found on the left side of the oven. Additionally, there is a **bowl** located on the left side of the kitchen, and a **bottle** can be seen on the right side.

**After:** The image depicts a kitchen with an old-fashioned white **stove** top **oven**. The stove has four burners, with two on the left side and two on the right side. Above the stove, there is a light that illuminates the cooking area. The kitchen is well-equipped with various utensils and appliances. There are two **knives**, one on the left side of the stove and the other on the right side. Additionally, there are two **spoons**, one near the left knife and the other near the right knife. A **bottle** can be seen on the left side of the stove, and a **clock** is mounted on the wall above the oven.

Figure 16: **Qualitative results for LLaVA hallucination intervention**. Our algorithm removes hallucinations and, at times, adds correctly detected objects.

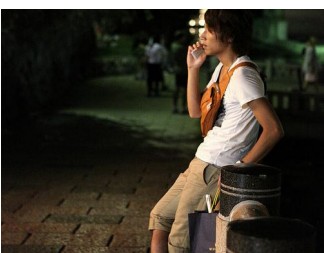

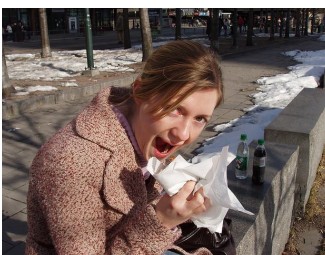

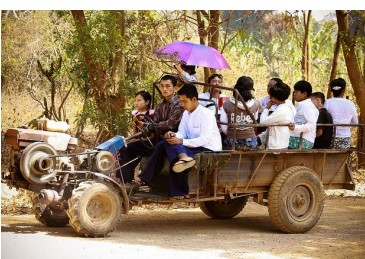

**Before:** The image depicts a young **woman** sitting on a **bench**, leaning against a wall, and talking on her **cell phone.** She is wearing a white shirt and appears to be engrossed in her conversation. In the background, there are several other people walking around, some of them carrying bags or **purses.** The scene takes place at night, with **streetlights** illuminating the area and casting shadows on the surroundings.

**After:** A young **man** is sitting on a railing, leaning against it while talking on his **cell phone**. He is wearing a **backpack** and appears to be engrossed in his conversation. There are several other people in the scene, some of whom are standing or walking around. One person is closer to the man on the railing, while others are scattered throughout the area. A **handbag** can be seen nearby, possibly belonging to one of the people in the scene.

**Before:** The image features a young **woman** sitting on a **bench** outside, eating a **hot dog** wrapped in a napkin. She appears to be enjoying her food as she takes a big bite out of the hot dog. There are several **bottles** scattered around the scene, including one close to the woman's left hand and another near her right foot. In addition to the bottles, there are two **cars** visible in the background, one on the left side and another on the right side of the image.

**After:** The image depicts a young **woman** sitting on a **bench** outside, eating a **hot dog** wrapped in a napkin. She appears to be enjoying her meal while surrounded by other **people** who are walking around the area. There are several **bottles** scattered throughout the scene, likely containing drinks or other beverages. In the background, there is a building with multiple windows visible.

**Before:** The image depicts a group of **people** riding in an old-fashioned, open-air **truck.** The truck is carrying a large number of people, with some sitting on the bed of the truck and others standing around it. There are several **umbrellas** in the scene, one of which is held by a person in the back of the truck, providing shade for the passengers. In addition to the umbrella, there are several **handbags** scattered throughout the scene, likely belonging to the passengers. The truck appears to be making its way through a wooded area, surrounded by trees and foliage.

**After**: The image features a group of **people** riding in an old-fashioned, open-air vehicle, likely an old tractor or **truck**, on a dirt road surrounded by trees. They are enjoying a leisurely ride through the countryside, with one person holding a purple **umbrella** to protect themselves from the sun. There are several people in the vehicle, with some seated in the front and others in the back. The scene captures the essence of a relaxed and enjoyable day out in the countryside.

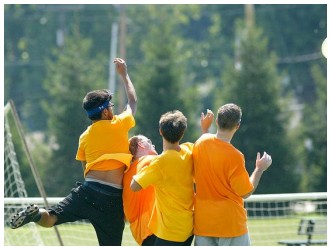

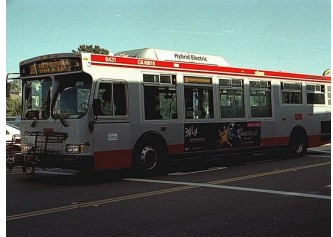

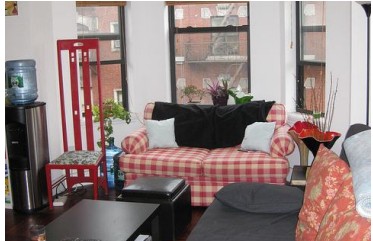

**Before:** The image depicts a group of four young **men** playing a game of **frisbee** on a grassy field. They are all wearing orange shirts, and one of them is jumping up to catch the frisbee as it flies through the air. There are several **benches** scattered around the field, providing seating options for the players and spectators. In addition to the benches, there are two **backpacks** visible in the scene, likely belonging to the players or spectators. Overall, the scene captures a lively and energetic game of frisbee among friends.

**After:** There is a group of young **men** playing a game of **frisbee** on a soccer field. They are all wearing matching shirts, and one of them is jumping up to catch the frisbee as it flies through the air. The other players are also actively participating in the game, trying to catch the frisbee or block their opponents' attempts to score. There are several **benches** scattered around the field, providing seating for those watching the game or taking a break from the action.

**Before:** The image depicts a city street with a red and white public transit **bus** parked on the side of the road. The bus is surrounded by other vehicles, including **cars**, **trucks**, and a **motorcycle**. There are several **people** visible in the scene, some standing near the bus and others walking along the sidewalk. A **bicycle** is also present in the scene, likely belonging to one of the pedestrians. Overall, the scene showcases a busy urban environment with various modes of transportation in use.

**After**: A red and white city **bus** is parked on the side of a street, waiting for passengers to board. There are several **people** in the vicinity of the bus, including one person standing near the front of the bus, another person walking towards the bus, and a third person further away from the bus. A **bicycle** is also visible in the scene, likely belonging to one of the pedestrians. In the background, there is a **car** parked on the other side of the street.

**Before:** The image depicts a cozy living room filled with various furniture and decorations. There is a black **couch** placed in the center of the room, surrounded by several **potted plants**. On the left side of the room, there is a coffee table with two **cups** placed on it. A **television** can be seen on the right side of the room, positioned close to the wall. In addition to the couch and coffee **table**, there are several potted plants scattered throughout the room, adding a touch of greenery to the space. Two **bottles** are also visible, one on the left side of the room and the other on the right side, near the coffee table. Overall, the living room has a warm and inviting atmosphere, perfect for relaxing and spending time with friends or family.

**After:** The image depicts a well-appointed living room with a **couch**, coffee table, and a **dining table**. There are several **potted plants** scattered throughout the room, adding a touch of greenery to the space. A bookshelf is visible on the left side of the room, showcasing various **books** and decorative items. In the center of the room, there is a coffee table surrounded by **chairs**, providing a comfortable seating area for guests. The room has a cozy and inviting atmosphere, perfect for relaxation and social gatherings.

Figure 17: **Qualitative results for InstructBLIP hallucination intervention**.

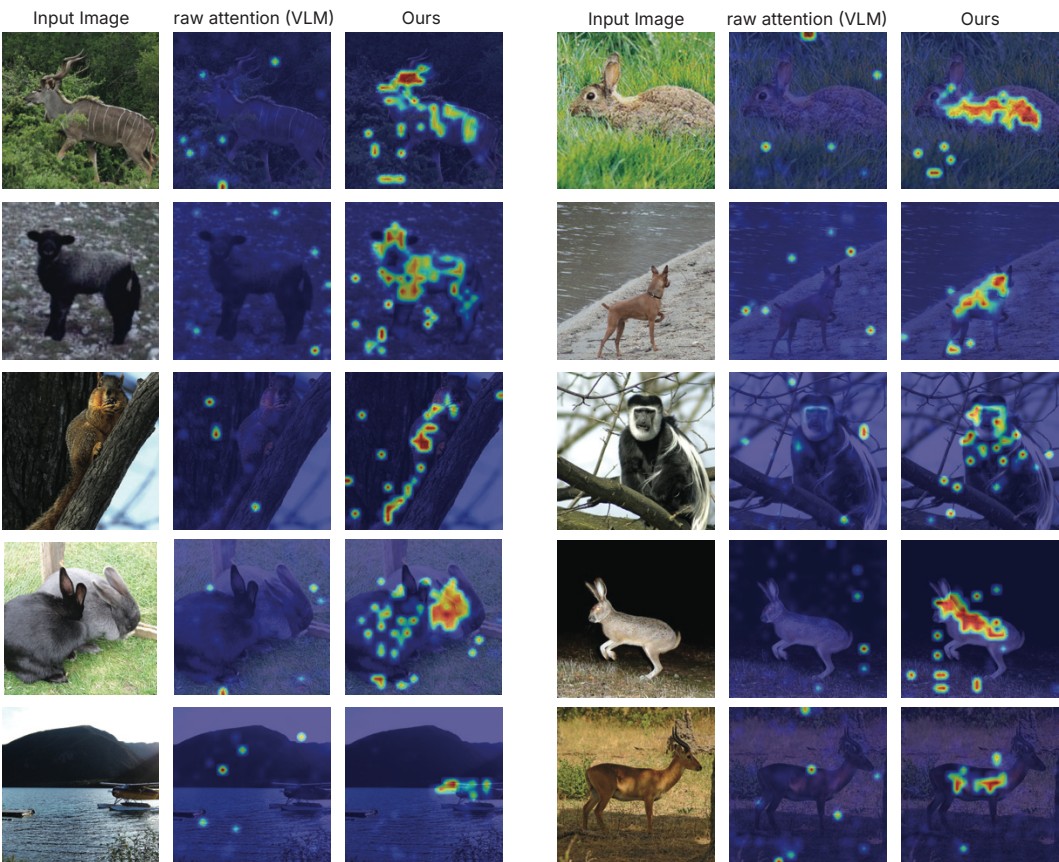

Figure 18: **Object Localization Samples**.

| Class | T3% | T5% | T10% | Patches | Class | T3% | T5% | T10% | Patches |
|---|---|---|---|---|---|---|---|---|---|
| airplane | 51.0 | 61.8 | 72.5 | 102 | kite | 66.7 | 77.8 | 88.9 | 9 |
| apple | 70.2 | 76.6 | 80.9 | 47 | knife | 24.0 | 26.0 | 34.0 | 50 |
| backpack | 44.8 | 48.5 | 58.5 | 614 | laptop | 34.1 | 38.3 | 46.1 | 334 |
| banana | 77.2 | 79.2 | 83.2 | 101 | microwave | 23.8 | 37.2 | 55.2 | 223 |
| baseball bat | 0.0 | 33.3 | 33.3 | 3 | motorcycle | 64.7 | 70.6 | 78.3 | 984 |
| baseball glove | 65.4 | 69.2 | 73.1 | 26 | mouse | 50.0 | 50.0 | 66.7 | 6 |
| bear | 37.6 | 41.4 | 47.6 | 739 | orange | 44.2 | 50.0 | 57.7 | 52 |
| bed | 44.0 | 47.8 | 52.0 | 2373 | oven | 38.9 | 54.4 | 78.9 | 507 |
| bench | 61.3 | 63.9 | 66.6 | 524 | parking meter | 20.9 | 24.8 | 29.6 | 230 |
| bicycle | 27.2 | 32.3 | 58.3 | 235 | person | 0.5 | 1.1 | 12.6 | 11528 |
| bird | 75.5 | 79.4 | 80.6 | 155 | pizza | 41.8 | 52.1 | 69.4 | 3146 |
| boat | 26.2 | 29.7 | 38.8 | 516 | potted plant | 43.8 | 52.1 | 63.0 | 192 |
| book | 28.1 | 39.0 | 44.8 | 210 | refrigerator | 23.7 | 31.4 | 50.0 | 156 |
| bottle | 46.2 | 53.4 | 62.0 | 208 | remote | 14.3 | 17.3 | 17.3 | 98 |
| bowl | 22.1 | 25.0 | 31.9 | 1364 | sandwich | 40.2 | 44.4 | 54.3 | 468 |
| broccoli | 75.9 | 77.2 | 79.7 | 79 | scissors | 71.9 | 71.9 | 71.9 | 32 |
| bus | 49.8 | 54.1 | 61.3 | 1786 | sheep | 49.5 | 52.6 | 56.6 | 489 |
| cake | 43.2 | 51.4 | 83.5 | 1182 | sink | 57.0 | 60.3 | 65.1 | 272 |
| car | 29.4 | 37.7 | 52.7 | 714 | skateboard | 48.1 | 50.6 | 57.7 | 239 |
| carrot | 50.0 | 50.0 | 75.0 | 4 | skis | 27.6 | 34.2 | 44.7 | 76 |
| cat | 57.7 | 61.5 | 66.5 | 2239 | snowboard | 37.5 | 41.7 | 47.9 | 48 |
| cell phone | 73.4 | 76.6 | 82.8 | 64 | spoon | 35.5 | 43.5 | 53.2 | 62 |
| chair | 31.6 | 33.1 | 37.6 | 516 | sports ball | 4.4 | 8.9 | 20.0 | 45 |
| clock | 67.6 | 70.6 | 75.9 | 299 | stop sign | 85.7 | 88.6 | 89.9 | 237 |
| couch | 33.4 | 38.9 | 63.2 | 2435 | suitcase | 38.1 | 40.7 | 44.7 | 472 |
| cow | 51.9 | 58.6 | 67.0 | 324 | surfboard | 48.6 | 57.5 | 69.9 | 146 |
| cup | 9.4 | 14.9 | 30.4 | 181 | teddy bear | 38.5 | 43.1 | 47.7 | 239 |
| dining table | 25.4 | 47.6 | 74.9 | 7403 | tennis racket | 88.9 | 88.9 | 88.9 | 9 |
| dog | 45.9 | 51.3 | 59.7 | 1057 | tie | 46.2 | 49.7 | 52.8 | 197 |
| donut | 34.8 | 40.0 | 45.2 | 115 | toilet | 92.0 | 97.3 | 99.6 | 1131 |
| elephant | 47.0 | 57.4 | 64.9 | 902 | toothbrush | 78.9 | 78.9 | 100.0 | 19 |
| fire hydrant | 43.4 | 47.0 | 50.1 | 419 | traffic light | 45.5 | 45.5 | 45.5 | 11 |
| fork | 41.4 | 45.7 | 54.3 | 70 | train | 40.7 | 45.0 | 52.1 | 3008 |
| frisbee | 50.0 | 60.6 | 71.2 | 66 | truck | 34.5 | 40.1 | 54.6 | 930 |
| giraffe | 32.0 | 39.8 | 62.8 | 810 | tv | 27.2 | 32.6 | 37.6 | 298 |
| handbag | 42.2 | 53.0 | 57.8 | 83 | umbrella | 39.5 | 40.5 | 42.8 | 526 |
| horse | 63.7 | 65.8 | 68.8 | 240 | vase | 29.9 | 32.6 | 45.1 | 264 |
| hot dog | 56.9 | 62.7 | 69.3 | 394 | wine glass | 61.9 | 67.0 | 74.3 | 218 |
| keyboard | 43.1 | 47.7 | 49.2 | 65 | zebra | 33.0 | 35.9 | 44.5 | 373 |
| **Overall** | **32.7** | **39.3** | **52.2** | **55988** | | | | | |

Table 8: **Per-class patch classification accuracy.** For each COCO class, we show the percentage of patches containing that object that top-k logit lens predictions can correctly identify.

