# OpenReview forum: "Interpreting and Editing Vision-Language Representations to Mitigate Hallucinations"
_ICLR.cc/2025/Conference — ICLR 2025 Poster_

### Official Review · Reviewer_LL1m · 2024-11-02

**Soundness:** 3
**Presentation:** 3
**Contribution:** 2
**Rating:** 5
**Confidence:** 5

**Summary:**

The paper explores the internal representations of Vision-Language Models (VLMs) to address the persistent issue of hallucinations. The authors project VLMs' internal image representations onto their language vocabulary to identify differences in token output probabilities between real and hallucinated objects. They introduce a knowledge erasure algorithm, PROJECTAWAY, which removes hallucinations by linearly orthogonalizing image features with respect to hallucinated object features. The study demonstrates that targeted edits to a model's latent representations can reduce hallucinations while preserving performance. Additionally, the paper presents a method for zero-shot segmentation using the logit lens technique, showing comparable performance to state-of-the-art methods.

**Strengths:**

- The paper presents a newmethod for reducing object hallucinations in VLMs by editing their latent representations and the introduction of PROJECTAWAY offers a new technique for selectively removing hallucinated objects from VLMs' outputs.

- The authors provide a thorough analysis of the internal confidence values for object presence and absence, offering empirical evidence that supports their claims.

**Weaknesses:**

- While the paper focuses on object hallucinations, it does not explore the applicability of the methods to other elements of visual scenes, such as people, attributes, or actions. The editing approach may struggle with abstract or complex sentences involving object attributes or interactions, which are not explicitly addressed in the paper.

- Could the authors elaborate on the potential impact of their editing techniques on other aspects of model performance, such as accuracy in non-hallucination tasks?

- The paper's reliance on LLaVA and InstructBLIP as baseline MLLMs does not provide a comprehensive comparison with the latest state-of-the-art models.

**Questions:**

-  Would the authors consider including comparisons with the latest MLLMs, such as those incorporating more advanced architectures or larger datasets, to validate the robustness of their approach?

---

> ### Author Response · Authors · 2024-11-16
>
> We thank the reviewer for the valuable comments and feedback on our paper and address the concerns below.
>
> ### Applicability of the methods to other elements of visual scenes (ex. people, attributes, actions)
>
> We focus on object hallucinations because they have standard evaluation suits, and it is difficult to get precise quantitative results for attribute hallucinations (mostly due to various possible phrasings). However, in Appendix A.7, we’ve added qualitative examples for attribute hallucinations (color, object number) based on images and questions from the VQA 2.0 challenge [1]. We find that our model confidence scores (Section 3) can identify when responses are inaccurate, and our editing technique can correct them appropriately.
>
> ### Potential impact of editing on accuracy in non-hallucination tasks
>
> Our analysis examines overall caption quality (not just tied with hallucinations) by measuring changes in correctly detected (CD) objects – the non-hallucinated objects that appear in the scene. As Table 1 shows, our editing technique does not produce a substantial change in CD objects, indicating that the new captions convey a similar degree of specificity for the objects contained within the scene. Appendix A.7 also shows the potential for using our editing technique to improve VQA performance by reducing attribute hallucinations. Outside of editing, we further show promising results for using logit lens to perform zero-shot classification, another non-hallucination task, in Appendix A.8.
>
> ### Other state-of-the-art MLLMs
> As Reviewer bJVo03 mentions, “InstructBLIP and LLaVA are representative LVLMs,” but we agree that the landscape of LVLMs is constantly evolving and want to ensure our analysis is thorough. We conducted additional evaluations on the same 500-image validation subset from Section 5.2 (“Hallucination reduction”) using more recent models – LLaVA-NEXT 7B [2] and Cambrian-1 8B [3] with Llama 3. The results demonstrate consistency with our original findings, suggesting that our conclusions generalize across model architectures. Our method results in a 27.73% reduction in hallucinations with LLaVA-NEXT and 28.26% with Cambrian-1. For simplicity, we use the same hyperparameters for LLaVA, though optimizing this selection would likely result in further improvements. While our original baselines are LLaVA and InstructBLIP, which are thoroughly evaluated in related hallucination reduction papers [4], this supplementary evaluation strengthens our claims. We include these new results in Appendix A.5.
>
> [1] https://visualqa.org/index.html
>
> [2] https://llava-vl.github.io/blog/2024-01-30-llava-next/
>
> [3] https://arxiv.org/abs/2406.16860
>
> [4] https://arxiv.org/pdf/2311.17911

---

> > ### Author Response · Authors · 2024-11-23
> >
> > Dear Reviewer,
> >
> > Thank you for taking the time to review our paper and provide valuable feedback. As the discussion phase is nearing its conclusion and there will be no second stage of author-reviewer interactions, we would like to confirm if our responses from a few days ago have effectively addressed your concerns. We hope they have resolved the issues you raised. If you require further clarification or have additional questions, please don’t hesitate to reach out. We are happy to continue the conversation.
> >
> > Thank you,
> >
> > The authors

---

> > > ### Comment · Reviewer_LL1m · 2024-11-26
> > > **response to author.**
> > >
> > > Thank you for your detailed response to my comments. Your clarifications have addressed many of my concerns, and I am pleased to update my score to 5.

---

> > > > ### Author Response · Authors · 2024-11-26
> > > >
> > > > We thank the reviewer for acknowledging our rebuttal. We would like to ask if the reviewer has any other concerns that can be addressed by us.
> > > >
> > > > Thank you,
> > > >
> > > > The authors

---

### Official Review · Reviewer_4Zea · 2024-11-03

**Soundness:** 3
**Presentation:** 2
**Contribution:** 3
**Rating:** 6
**Confidence:** 4

**Summary:**

This paper presents a novel approach to understanding and editing vision-language models' (VLMs) internal representations through vocabulary projection and linear orthogonalization. By introducing a knowledge erasure algorithm PROJECTAWAY, the authors demonstrate significant improvements in hallucination reduction (up to 25.7%) and achieve competitive performance in zero-shot segmentation, while providing new insights into how VLMs process visual information.

**Strengths:**

1. The paper presents a novel approach to interpreting and editing VLM representations through vocabulary projection and linear orthogonalization, requiring no model retraining or external components.
2. The work provides insights into VLM behavior by revealing the relationship between internal confidence scores and object presence.

**Weaknesses:**

1. The paper's main analysis and evaluations (Sections 3 and 4) are predominantly conducted under the assumption that hallucinated objects are known beforehand using ground truth annotations. While Section 5 addresses this limitation with a more realistic approach using internal confidence thresholds, this should have been the primary evaluation framework. The current structure potentially overestimates the method's effectiveness by evaluating under idealized conditions.
2. The paper's structure is suboptimal, with the main analysis focusing on scenarios using ground truth annotations while relegating the more realistic approach to the applications section.
3. The choice to use the last token for multi-token object representations (e.g., "hot dog", "dining table") lacks sufficient justification and empirical validation. The paper does not analyze potential issues with this approach, such as cases where the last token might not be the most semantically meaningful (e.g., "traffic light" where "light" alone might be ambiguous) or how this choice affects the method's performance compared to alternatives like averaging all tokens or using the first token.

**Questions:**

1. The paper uses the model's unembedding matrix to interpret intermediate layer representations, but this matrix is trained for the final output layer. Have you conducted any layerwise probing or training of separate unembedding matrices for intermediate layers? This could affect the reliability of interpreting earlier layer representations.

---

> ### Author Response · Authors · 2024-11-16
>
> We thank the reviewer for giving feedback on the paper and providing valuable comments. We address the concerns and questions below.
>
> ### The current structure performs its main analysis of the editing technique under idealized conditions and potentially overestimates its effectiveness as a result.
>
> We see the reviewer’s point about highlighting the practical approach upfront. Our intention in Section 4 (“Erasing Knowledge from VLMs”) is to study the editing technique’s effects independent of model confidences. This section highlights the surprising result that linear orthogonalization can effectively remove knowledge of objects from image captions, whether they are hallucinated or not. In our intro and abstract, we only highlight the hallucination reduction results from Section 5.2 (“Hallucination reduction”) to avoid suggesting that the idealized findings from Section 4 represent expected outcomes in practical applications. Nevertheless, we would be happy to hear suggestions for a better structure from the reviewers.
>
> ### “Have you conducted layerwise probing or training of separate unembedding matrices?”
> We use the model’s unembedding matrix to interpret intermediate layer representations and provide a training-free method for interpreting the internal representations. The logit lens [1] method on text-only models shows that the model’s unembedding matrix effectively interprets LLMs, and we demonstrate that, surprisingly, it is true for LVLMs as well. We intend to show that it is possible to interpret the internals of these models without requiring additional training and present a novel interpretability method that can be widely used across VLMs without repeated training.
>
> ### Justifying Last Tokens for Multi-Token Object Representations
> Our approach of using the last tokens is motivated by past work that find that information about multi-token entities is moved to the last token position. For example, [2] finds that the last subject token encodes crucial factual associations, and [3] demonstrates that information is carried over to the last token position through relation propagation and attribute extraction. Thus, extracting a residual hidden representation of the last token, which is conditioned on the previous tokens of the class, is the most likely to contain the concept of the whole class (ex. “traffic light”) and not merely a single part.
>
> [1] https://www.lesswrong.com/posts/AcKRB8wDpdaN6v6ru
>
> [2] https://rome.baulab.info/
>
> [3] https://arxiv.org/abs/2304.14767

---

> > ### Author Response · Authors · 2024-11-23
> >
> > Dear Reviewer,
> >
> > Thank you for taking the time to review our paper and provide valuable feedback. As the discussion phase is nearing its conclusion and there will be no second stage of author-reviewer interactions, we would like to confirm if our responses from a few days ago have effectively addressed your concerns. We hope they have resolved the issues you raised. If you require further clarification or have additional questions, please don’t hesitate to reach out. We are happy to continue the conversation.
> >
> > Thank you,
> >
> > The authors

---

> > > ### Author Response · Authors · 2024-11-27
> > >
> > > Dear Reviewer,
> > > Thank you again for your valuable comments. We would like to ask once again if our rebuttal addressed your concerns and if there is anything else that can resolve the issues you raised.
> > >
> > > Thank you,
> > >
> > > The authors

---

> > > > ### Author Response · Authors · 2024-11-30
> > > >
> > > > Dear Reviewer,
> > > >
> > > > We appreciate the valuable feedback you have given. As the discussion window will be closing in a few days, we would like to ask again if our rebuttal has addressed your concerns and if there are any remaining problems we can address.
> > > >
> > > > Thank you,
> > > >
> > > > The authors

---

### Official Review · Reviewer_bJVo · 2024-11-04

**Soundness:** 4
**Presentation:** 4
**Contribution:** 3
**Rating:** 8
**Confidence:** 5

**Summary:**

The authors use Logit Lens to interpret the intermediate image representations in LVLMs. For a given image embedding, they extract the latent representation of the image embedding at a specific layer, taking the logit lens to get the probability distribution over the vocabulary.
- The highest probability of an object across image representations and layers, can act as the internal confidence of VLMs. The confidences for objects present are significantly higher than those of objects not present in the image.
- The authors propose an algorithm, ProjectAway, erasing objects from image representations.
- Moreover, they find that, using the internal confidence values, they can localize the objects in the image patches.

The authors show three applications of their findings and the algorithm: hallucination detection, hallucination mitigation, and zero-shot segmentation.

**Strengths:**

- The findings are well-written and easy to understand.
- The experiments are comprehensive, exploring different aspects of internal visual information and covering different tasks.
- The proposed approach achieves significant improvements or comparable performance to SoTA on three applications.

**Weaknesses:**

### Major
- Is the unembedding matrix for image representations directly from the LVLM last layer, or trained by the authors?
- Previous papers report the modality gap between language and vision in VLMs. In my experiments, I also notice that the distribution of vision tokens are significantly different from that of textual tokens. So I’m surprised that the logit lens can be directly used in image representations.
I’m curious about the classification accuracy of logit lens. For example, if we feed a patch of cat, how accurate is the logit lens method to identify it is cat.
- Lines 200-202, the authors “randomly sample a subset of” objects not present. I’m wondering if this random sampling will choose some objects “obviously” not present in the image, making the comparison of the internal confidence too easy. It might be better if the authors can show: the confidence distribution of objects that commonly appear with objects in the image but not present this time.
- Section 5.3, I think LLaVA tends to generate some very general class when classifying an image, like predicting "dog" instead of “husky”. Are the authors using the generated class name from LLaVA no matter what it is or using the ground truth label?

### Minor
- InstructBLIP and LLaVA are representative LVLMs, but recent LVLMs are using more complicated vision embedding techniques [1, 2]. I’m wondering if the proposed method can still work with these new architectures.
- If we want to detect or remove the hallucinated objects, the propose method needs to know the object name. I'm wondering if the proposed method can work on a popular hallucination benchmark POPE [3]? In POPE, every sample is a "yes or no" question, like "Is there a person in the image?"
- Other limitations like handling multi-token classes have been mentioned in the paper.

[1] LLaVA-NeXT. https://llava-vl.github.io/blog/2024-01-30-llava-next/

[2] Cambrian-1: A Fully Open, Vision-Centric Exploration of Multimodal LLMs https://arxiv.org/abs/2406.16860

[3] Evaluating Object Hallucination in Large Vision-Language Models. https://arxiv.org/abs/2305.10355

**Questions:**

Please see the Weaknesses section.

---

> ### Author Response · Authors · 2024-11-16
>
> We thank the reviewer for the detailed questions and feedback and aim to address them below.
>
> ### “Is the unembedding matrix trained by the authors?”
> We use the unembedding matrix from the LVLM and do not train, as we intend to provide a training-free method for interpreting the internal representations of LVLMs. The logit lens method, applied to text-only models, showed that the model’s unembedding matrix effectively interprets language models. We show that this capability can surprisingly be extended to LVLMs.
>
> ### Vision-language modality gap
> We agree with the important point about the modality gap and token distributions. We include classification results in Table 8 in the Appendix, performing patch-level evaluation on 500 images in the COCO dataset. We find that the classification accuracy varies highly between different COCO classes, with strong top-3 classification accuracy with classes such as “toothbrush” (78.9%), “toilet” (92%), and “banana” (77.2%) and much lower accuracy with some classes such as “person” (0.5%) and cup (9.4%). We hypothesize that the variation stems from how consistently objects are represented linguistically - classes that map to specific, consistent tokens perform better than those that can be described with many specific terms (e.g., "person" → "doctor", "skier", "girl"). The LVLM sometimes captions the image with more specific terms (such as in the case of class “person”) so we can interpret the image representations with these more specific terms well, e.g. since there is only one way to describe “banana”. More importantly, our quantitative results demonstrate that the logit lens effectively captures the model’s learned semantic alignments in practice. Section 5.1 shows that our method can distinguish objects present vs. not present, achieving significant improvements in hallucination detection, despite this modality gap. Additionally, in our zero-shot segmentation results (Section 5.3), we demonstrate that these projections accurately localize classes spatially. These quantitative results across multiple tasks suggest that the logit lens captures the model’s internal understanding of visual semantics, specifically with practical applications in hallucination intervention, despite modality differences.
>
> ### Does random sampling choose some objects “obviously” not present in the image?
> We sample only from the set of 80 COCO classes, where many objects commonly co-occur, and believe that the strong performance across applications validates that distributions of objects present and not present are reliably separable. In Section 5.1 (“Hallucination detection”), we narrow down the scope of randomly selected objects to only hallucinated objects, where hallucinations are often objects that are not present in the image but less obviously so. Through this specific application, we intend to show here that the logit lens can classify these objects as present or not present with strong results (mAP improvement by 22.45% in LLaVA and 47.17% in InstructBLIP), even while these objects not being present may be less obvious.
>
> ### Last Tokens for Multi-Token Object Representations
> Our approach of using the last tokens is motivated by past work that find that information about multi-token entities is moved to the last token position. For example, [1] finds that the last subject token encodes crucial factual associations, and [2] demonstrates that information is carried over to the last token position through relation propagation and attribute extraction. Thus, extracting a residual hidden representation of the last token, which is conditioned on the previous tokens of the class, is the most likely to contain the concept of the whole class (ex. “traffic light”) and not merely a single part.
>
> ### “Is the generated class name from LLaVA or the ground truth label used?
> Similar to the reviewer’s findings, we found that LLaVA tends to generate some very general class when classifying an image. We use the generated class name from LLaVA in zero-shot segmentation for two reasons. (1) We generate the segmentation without knowing the ground truth label, and only the LLaVA object prediction, to have true end-to-end zero-shot segmentation. (2) If LLaVA predicts a “dog” rather than a “husky” in the image, we find that it maps the image representations closer to “dog” tokens than to “husky” tokens. This is likely because this is how it internally processes the objects in the image representations (as “dog,” not “husky,” resulting in higher internal confidence for “dog”), which we interpret with text.
>
> [1] https://rome.baulab.info/
>
> [2] https://arxiv.org/abs/2304.14767

---

> > ### Author Response · Authors · 2024-11-16
> >
> > ...Continuing the previous response:
> >
> > ### Newer Architectures
> > We appreciate the reviewer's observation about evolving LVLM architectures. We conducted additional evaluations on the same 500-image validation subset for LLaVA from Section 5.2 using more recent models, LLaVA-NEXT 7B and Cambrian-1 8B with Llama 3. The results demonstrate consistency with our original findings, suggesting that our conclusions generalize across model architectures. Our method results in a 27.73% reduction in hallucinations with LLaVA-NEXT and 28.26% with Cambrian-1, where we empirically chose hyperparameters for editing. We include these new results in Appendix A.5.
> >
> > ### Evaluating on POPE
> > We do not use POPE in our evaluation because our editing technique is designed to remove the knowledge of objects or visual features from the image representations, not “yes” or “no”. However, in Appendix A.7, we added qualitative examples for questions from a VQA challenge, demonstrating that even for questions with short answers, our model confidence scores (Section 3) can detect inaccuracies and our editing technique can correct them.

---

> > > ### Comment · Reviewer_bJVo · 2024-11-16
> > >
> > > Thank you for your detailed response and additional experiments! It is a good paper and I would like to keep my score.

---

### Official Review · Reviewer_9Mwt · 2024-11-04

**Soundness:** 3
**Presentation:** 3
**Contribution:** 3
**Rating:** 5
**Confidence:** 4

**Summary:**

The paper addresses the issue of hallucinations in Vision-Language Models (VLMs) by interpreting and editing their internal representations. The authors apply the logit lens technique to project image representations onto the language vocabulary, discovering that objects present in the image have higher internal confidence scores compared to hallucinated objects. Utilizing this insight, they propose a method to detect hallucinations within VLMs. Furthermore, they introduce a knowledge erasure algorithm called PROJECTAWAY, which linearly orthogonalizes image features with respect to hallucinated object features to remove hallucinations from the model's output. The method is evaluated on two state-of-the-art VLMs, LLaVA 1.5 and InstructBLIP, showing a reduction in hallucinations by up to 25.7% on the COCO2014 dataset while preserving overall performance. Additionally, the authors demonstrate that their approach enables zero-shot segmentation by spatially localizing objects using internal confidence scores.

**Strengths:**

- The paper introduces a novel application of the logit lens technique to interpret the internal image representations of VLMs, providing new insights into how these models process visual information.
- The proposed knowledge erasure algorithm, PROJECTAWAY, is a simple yet effective method that relies solely on manipulating the internal features of the VLMs without requiring additional training or external modules.
- The approach enables zero-shot segmentation by leveraging internal confidence scores to spatially localize objects
- The paper seems clear and well-written.

**Weaknesses:**

- The proposed method requires specifying weight factors and selecting specific layers to retrieve text representations and apply edits. These hyperparameters are determined through ablation studies and do vary between models, and likely between datasets as well, requiring cumbersome ablation process to find good numbers.
- The experiments focus primarily on object hallucinations in image captioning tasks. It is unclear how the method performs on other types of hallucinations (e.g., action or attribute hallucinations) or on other tasks such as visual question answering (VQA).
- The impact of the method on overall caption quality is not thoroughly evaluated quantitatively. While the authors mention that the method preserves performance and provide some qualitative examples, additional quantitative evaluations would be interesting to see.
- The authors only seem to test their model on COCO2014.

**Questions:**

- How sensitive is the proposed method to the selection of weight factors and layers across different models and datasets? Is there a way to generalize these hyperparameters or make the method more robust to their selection?
- How does the method perform on other tasks, such as visual question answering (VQA) or on other datasets beyond COCO2014? Have you considered testing the method on benchmarks like LLaVA Bench or MM-Vet?
- Is there a way to automate or simplify the selection of hyperparameters (e.g., layers, weight factors) to make the method more practical for real-world applications?

---

> ### Author Response · Authors · 2024-11-16
>
> We thank the reviewer for the valuable comments on the paper and address the concerns below.
>
> ### Hyperparameters are determined through a cumbersome ablation process
> Our ablations in Figure 6 show that for LLaVA, there is significant (>15%) hallucination reduction across the vast majority of hyperparameter selections. When testing our hallucination reduction method with two different VLMs, Cambrian and LLaVA-Next, we took similar hyperparameters and achieved strong results despite the lack of ablation studies. Moreover, all of the ablations can be done automatically as a one-time cost for a given model using a simple grid search.
>
> ### Applying method to other types of hallucinations and other tasks like VQA
> We focus on object hallucinations because they have standard evaluation suits, and it is difficult to get precise quantitative results for attribute hallucinations (mostly due to various possible phrasings). However, in Appendix A.7, we’ve added qualitative examples for attribute hallucinations (color, object number) based on images and questions from the VQA 2.0 challenge [1]. We find that our model confidence scores (Section 3) can identify when responses are inaccurate, and our editing technique can correct them appropriately.
>
> ### Overall caption quality was not evaluated quantitatively
> We quantitatively evaluate caption quality by measuring changes in correctly detected (CD) objects, the non-hallucinated objects that appear in the scene. As Table 1 shows, editing hallucinations out does not lead to a substantial change in CD objects. We are not familiar with strong, comprehensive evaluation criteria that can automatically measure the quality of non-object attributes (ex. Color, shape, relation) in captions and are not prone to breaking under small phrase changes.
>
> ### The authors only seem to test their model on COCO2014.
> In Appendix A.9, we add more qualitative examples of hallucination reduction on images from LLaVA-Bench [2] and find that they align with the strong results seen with COCO2014. We primarily use COCO2014 because the object hallucination metric CHAIR is tied to the dataset and drives our quantitative results. The images contained within COCO2014 are diverse, and we use separate image data to select hyperparameters (ex. Section 4.2 - “Ablations”) and to test the model (Section 5.1 and 5.2).
>
> [1] https://visualqa.org/index.html
>
> [2] https://huggingface.co/datasets/liuhaotian/llava-bench-in-the-wild

---

> > ### Author Response · Authors · 2024-11-23
> >
> > Dear Reviewer,
> >
> > Thank you for taking the time to review our paper and provide valuable feedback. As the discussion phase is nearing its conclusion and there will be no second stage of author-reviewer interactions, we would like to confirm if our responses from a few days ago have effectively addressed your concerns. We hope they have resolved the issues you raised. If you require further clarification or have additional questions, please don’t hesitate to reach out. We are happy to continue the conversation.
> >
> > Thank you,
> >
> > The authors

---

> > > ### Author Response · Authors · 2024-11-27
> > >
> > > Dear Reviewer,
> > > Thank you again for your valuable comments. We would like to ask once again if our rebuttal addressed your concerns and if there is anything else that can resolve the issues you raised.
> > >
> > > Thank you,
> > >
> > > The authors

---

> > > > ### Author Response · Authors · 2024-11-30
> > > >
> > > > Dear Reviewer,
> > > >
> > > > We appreciate the valuable feedback you have given. As the discussion window will be closing in a few days, we would like to ask again if our rebuttal has addressed your concerns and if there are any remaining problems we can address.
> > > >
> > > > Thank you,
> > > >
> > > > The authors

---

### Author Response · Authors · 2024-11-16
**To all reviewers**

We thank all the reviewers for providing valuable comments and feedback on our paper. The reviewers describe our application of the logit lens technique on VLMs to be a “novel approach” that provides “new insights on how these models process visual information” (4Zea03; 9Mwt03). They mention how our editing technique, ProjectAway, is a “simple yet effective method” that requires “no model retraining or external components”(9Mwt03; 4Zea03). Our methods produce 3 applications that “achieve significant improvements or comparable performance to SoTA” (bJVo03). The reviewers find our experiments to be “comprehensive…and covering different tasks” and a “thorough analysis of the internal confidence values for object presence and absence” (bJVo03; LL1m02). Many of the reviewers also praise the clarity of our findings, stating they are “well-written” and “easy to understand” (9Mwt03; bJVo03).

There are 3 common concerns the reviewers raised that we hope we addressed in this rebuttal:

### Evaluating our methods on more advanced, recent VLMs (LL1m02, bJVo03)

While LLaVA and InstructBLIP follow a similar architecture as most VLMs today, we conduct additional evaluations on more recent VLMs like LLaVA-NeXT and Cambrian-1, which are trained with more advanced techniques on better datasets. Our results in Appendix A.5 show that our editing technique, paired with model confidences, is able to significantly reduce hallucinations (>25%) consistent with our other results. They also demonstrate that our method is robust to different hyperparameter selections as we did not run ablation studies to optimize them.

### Applying our method beyond object hallucinations (9Mwt03, LL1m02)

A few reviewers wanted to see our method applied beyond object hallucinations, such as to attribute (color, relation, object number) hallucinations. As it is difficult to get precise quantitative numbers due to the lack of standard benchmarks for attribute hallucinations, we instead provide qualitative examples in Appendix A.7 from a VQA task to demonstrate that our method can accurately detect and correct wrong answers to attribute-related (ex. “What color is <blank>?”) questions.

### Lack of justification for using last tokens for multi-token text representations (4Zea03, bJVo03)

Our editing technique, ProjectAway, uses text embeddings pulled from the last token of multi-token objects, and a few reviewers wanted further justification for this design choice. We primarily use the last token because past works have found models tend to store information about multi-token entities in the last token for later use. For example, [1] finds that the last subject token encodes crucial factual associations, and [2] demonstrates that information is carried over to the last token position through relation propagation and attribute extraction.

We hope that our additional evaluations and results address the concerns of the reviewers. We will incorporate the feedback into our paper and would be happy to hear of any further ways to strengthen our claims.

[1] https://rome.baulab.info/

[2] https://arxiv.org/abs/2304.14767

---

### Author Response · Authors · 2024-11-21

We appreciate the time and valuable feedback provided by all the reviewers on our work. We are thankful that the paper has been positively received overall. As the discussion period is nearing its end, we kindly request that all reviewers confirm whether our rebuttal has addressed their concerns and allow us the chance to respond to any additional follow-up. Thank you once more for your participation.

---

### Meta-Review · Area_Chair_QyDk · 2024-12-20

**Metareview:**

This paper proposes an algorithm to erase spurious knowledge from VLMs. The algorithm, coined ProjectAway, relies on Logit Lens to remove information about objects from image representations. The proposed approach is evaluated on several applications: hallucination detection, hallucination removal, as well as zero-shot segmentation. While reviewers raised some concerns regarding the practical applicability of the proposed approach (too much manual work), and some concerns regarding the evaluation, the work constitutes a good piece of work in the space of VLM interpretability. For the above reason, despite borderline ratings, I recommend accepting this paper.

**Additional Comments On Reviewer Discussion:**

The authors provided a rebuttal adressing some of the reviewer's concerns. Nonetheless, 2/4 reviewers did not acknowledge the author's response (5 and 6). One reviewer updated their score from 3->5. Ratings went from 3568 to 5568.

---

### Decision · Program_Chairs · 2025-01-22

Accept (Poster)